# Multimodal Estimation of Sine Dwell Vibrational Responses from Aeroelastic Flutter Flight Tests

**Sami Abou-Kebeh** *,†, **Roberto Gil-Pita** †  and **Manuel Rosa-Zurera** †

Signal Theory and Communications Department, University of Alcalá, 28805 Alcalá de Henares, Spain; roberto.gil@uah.es (R.G.-P.); manuel.rosa@uah.es (M.R.-Z.)

* Correspondence: sami.a.k.ll@gmail.com
† These authors contributed equally to this work.

**Abstract:** Aircraft envelope expansion during new underwing stores installation is a challenging problem, mainly related to the aeroelastic flutter phenomenon. Aeroelastic models are usually very hard to model, and therefore flight tests are usually required to validate the aeroelastic model predictions, which given the catastrophic consequences of reaching the flutter point pose an important problem. This constraint favors using short time excitations like Sine Dwell to perform the flight tests, so that the aircraft stays close to the flutter point as little time as possible, but short time data implies a poor spectrum resolution and therefore leads to inaccurate and non repetitive results. The present paper will address the problem related to processing Sine Dwell signals from aeroelastic Flutter Flight Tests, characterized by very short data length (less than 5 s) and low frequency (less than 10 Hz) and used to identify the natural modes associated with the structure. In particular, a new robust technique, the PRESTO algorithm, will be presented and compared to a Matching Pursuit estimation based on Laplace Wavelet. Both techniques have demonstrated to be very accurate and robust procedures on very short time (Sine Dwell) signals, with the particularity that the Laplace Wavelet estimation has already been validated over F-18 real Flutter Flight Test data as described in different papers. However, the PRESTO algorithm improves the performance and accuracy of the Laplace Wavelet processing while keeping its robustness, both on real and simulated data.

**Keywords:** Flutter Flight Test; Matching Pursuit; robust flutter data processing; system identification; very short time signals; low frequency identification; damping identification; dwell excitations; wavelet; sine dwell



## 1. Introduction

The flight envelope of an aircraft is the locus where the aircraft can safely fly. If the inertial, elastic or aerodynamic characteristics of the aircraft change it is necessary to update and expand the flight envelope [1–3]. The flight envelope is defined by several boundaries, typically the aircraft ceiling, stall limit, maximum power and aeroelastic phenomena. An example of a typical flight envelope can be seen in Figure 1. In particular, one of the most important aeroelastic phenomena, the driver of the present paper, is the Aeroelastic Flutter, whose tests unfailingly imply "high" residual risk according to any standard risk analysis [4], since flutter flight conditions often end in catastrophic results. With this fact taken into account, extensive previous analyses must be performed on either new aircraft designs or old designs with substantial changes in the inertial/elastic/aerodynamic properties of the aerodynamic surfaces, typically due to the installation of new wing stores as required by different standards [5–7], since those are the factors that modify the flutter conditions. A Spanish Air Force F-18 fully configured and ready to perform a Flutter Flight Test, like the ones described in this paper, is depicted in Figure 2.

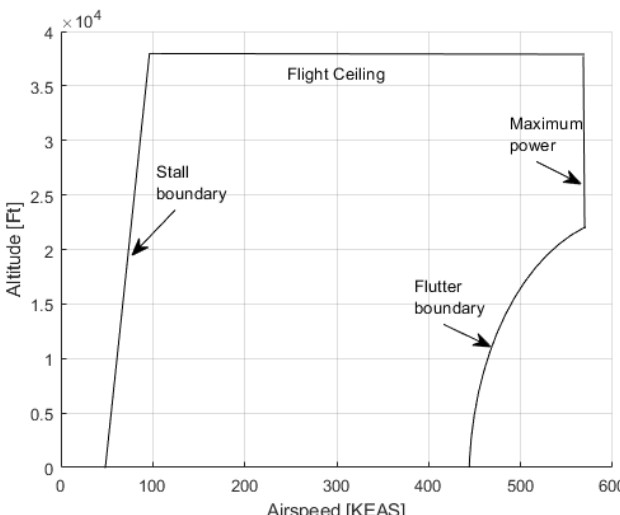

**Figure 1.** Depiction of a typical flight envelope. In this sample case the flutter boundary defines the low altitude-high airspeed right boundary, while the high altitude and high airspeed boundaries are defined by the wing surface and maximum available power. However, different aircraft may present different boundaries produced by other aeroelastic phenomena.

Aeroelastic flutter can be described as a self-excited vibration where two different natural modes of the structure, initially at different frequencies, start to modify their frequencies during flight as dynamic pressure increases, bringing them closer together. When the frequencies are close enough, they start to exchange energy producing a synchronized coupling, increasing the amplitude of at least one of the modes and producing the failure of the structure. The importance of damping is due to the fact that the damping factor of each mode changes as well. The mode whose amplitude increases faces an increase in damping factor (relative to an initial negative value), and when such damping factor passes from negative to positive values the amplitude vibration increases until the failure of the structure.

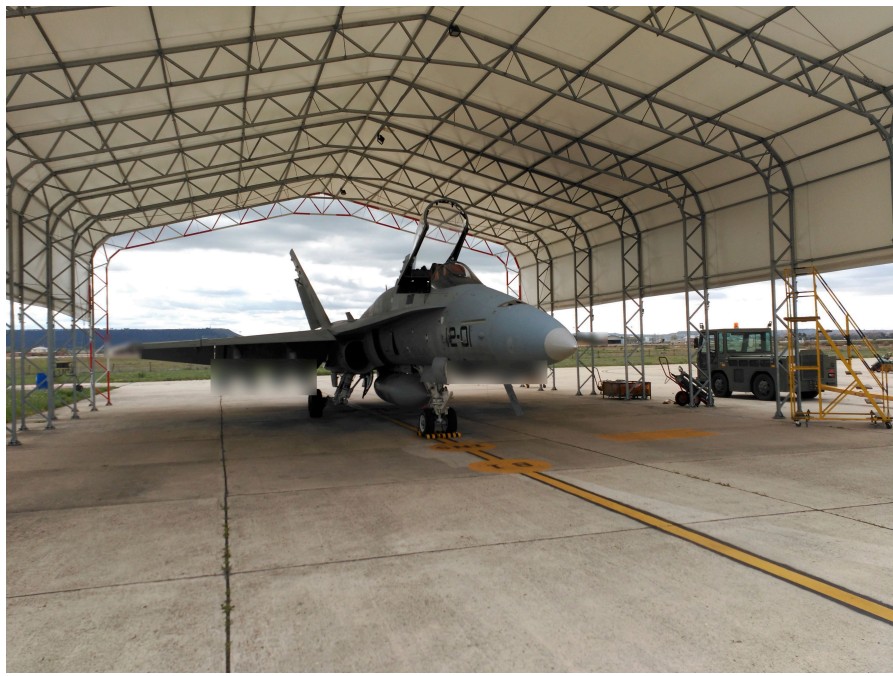

**Figure 2.** F-18 in the CLAEX platform ready to perform a Flutter Flight Test sortie. Stores configuration has been blurred due to confidentiality.

Nowadays one of the conditions to guarantee the airworthiness certification of any given aircraft is to demonstrate that it is free of flutter within the flight envelope [5,7,8]. Therefore, in the aforementioned cases an analysis process and Flutter Test must follow, starting by numerical methods to predict the flutter point (typically Nastran [9] and/or Zaero [10], although companies also use proprietary software to perform that analysis), ground vibration tests to validate the natural modes, frequencies and damping factors at zero airspeed (a good example is provided by [11]), and Flutter Flight Tests to validate the flight envelope. The whole process is out of the scope of this paper, although it is necessary to understand the whole process to acknowledge the importance and necessity of an accurate identification of the natural modes, frequencies and damping factors, and hence it will be briefly described in Section 2. As a reference, the process is documented by [1], and good examples are provided by [2,3].

Once the flight tests end and data are gathered, it is necessary to post-process data to characterize the aeroelastic behavior of the aircraft and validate the aeroelastic simulation model. In a general case there are many techniques described to process these data, like wavelet [12,13], non-linear techniques [14,15], Rational Fractional Polynomials [16–18], Prony and variants, like Matrix Pencil and Eigenvalue Realization Algorithm (ERA) [19–22], etc. Once the different frequencies and damping factors for the critical modes have been identified it is necessary to predict the flutter point (the corresponding envelope boundary) from extrapolation. Several methods are described to achieve this objective: [23–28] cite the most important, although they are not part of the scope of this paper.

The main two excitation techniques on the aircraft suited to gather Aeroelastic Flutter Flight Test data are Frequency Sweeps and Sine Dwells.

- **Frequency Sweeps** are chirp excitations from low to high frequency (values around 2 Hz to 20 Hz) and usually require 30 s to 60 s time signals. This is a big drawback when close to the flutter point, considering that the pilot needs to keep the aircraft in straight and level flight for such amount of time. However, they provide very good information when trying to characterize the system. Figure 3 shows a real Frequency Sweep from the gathered data, where the upper plot of the figure shows the aileron excitation and the lower plot of the figure shows the reading of one of the sensors.
- **Sine Dwells** consist of several excitations at a given frequency close to the expected natural frequency. In the case of this paper's datasets, the aircraft performs a battery of eight different excitations of 3 s each. For example, if the expected natural frequency lies at 4.5 Hz, the pilot will run eight different programs exciting the aircraft at 4.2 Hz, 4.3 Hz, etc. until 4.9 Hz, leaving approximately 3 s to 5 s relax time between each program to gather data. The main advantage of the Sine Dwell excitations is that the time on which the pilot flies under flutter conditions is very reduced, which considerably limits the risk associated to the tests. Figure 4 represents a Sine Dwell sample extracted from real Flutter Flight Test data. In this figure only two program runs are represented, while the full set consists of eight program runs. As in the case of the Frequency Sweep, the upper plot of the figure shows the aileron excitation and the lower plot the response of one of the sensors.

For the particular case of Sine Dwell excitations, characterized by being very short time series (3 s to 5 s) as related to the frequencies of interest (2 Hz to 10 Hz), not all the postprocessing techniques described above are suitable. Several require long times to get accurate results, usually requiring Frequency Sweeps lasting around 30 s to 60 s. For short time techniques, the most accurate identified are variants of Prony method [19–22], wavelet techniques [12,13], a bandwidth based estimation (Peak-Amplitude technique) proposed by [29] and the classical logarithmic decrements technique ([30], citing a generic text for the sake of completeness), although this last technique is limited to signals where only one single mode is present with close to zero phase angle. Note that the outcome of these techniques sometimes is not suitable for the estimation of multiple modes with close natural frequencies, as will be demonstrated in upcoming sections.

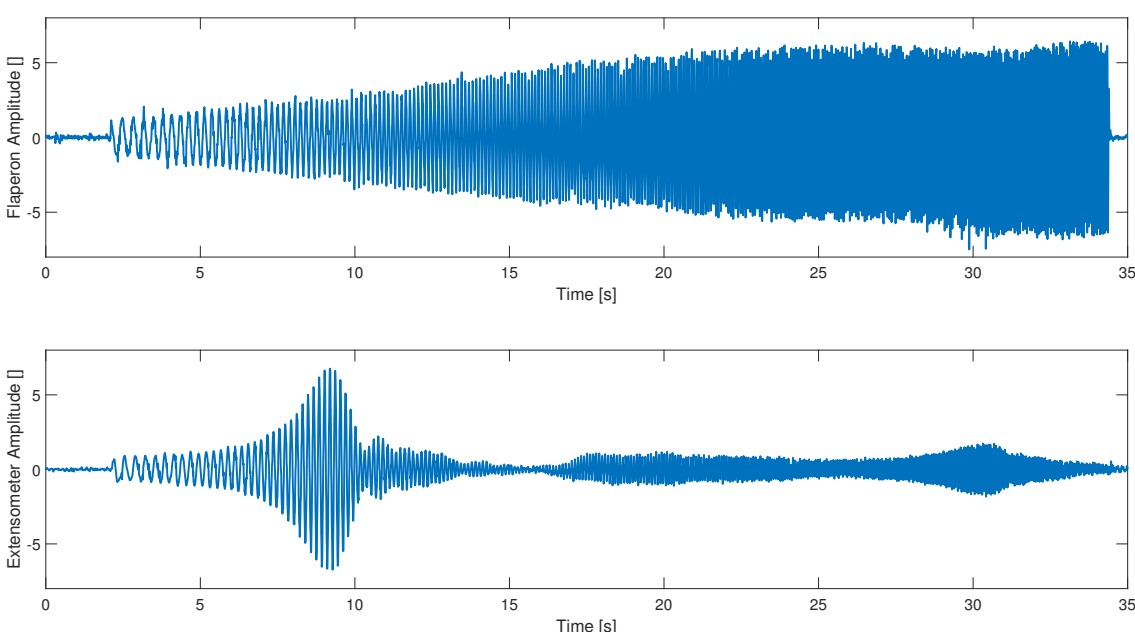

**Figure 3.** Frequency Sweep sample plot extracted from real Flutter Flight Test data. The upper plot shows the forced excitation on the flaperon through the Flight Controls, while the lower plot shows the reading from the wing extensometers.

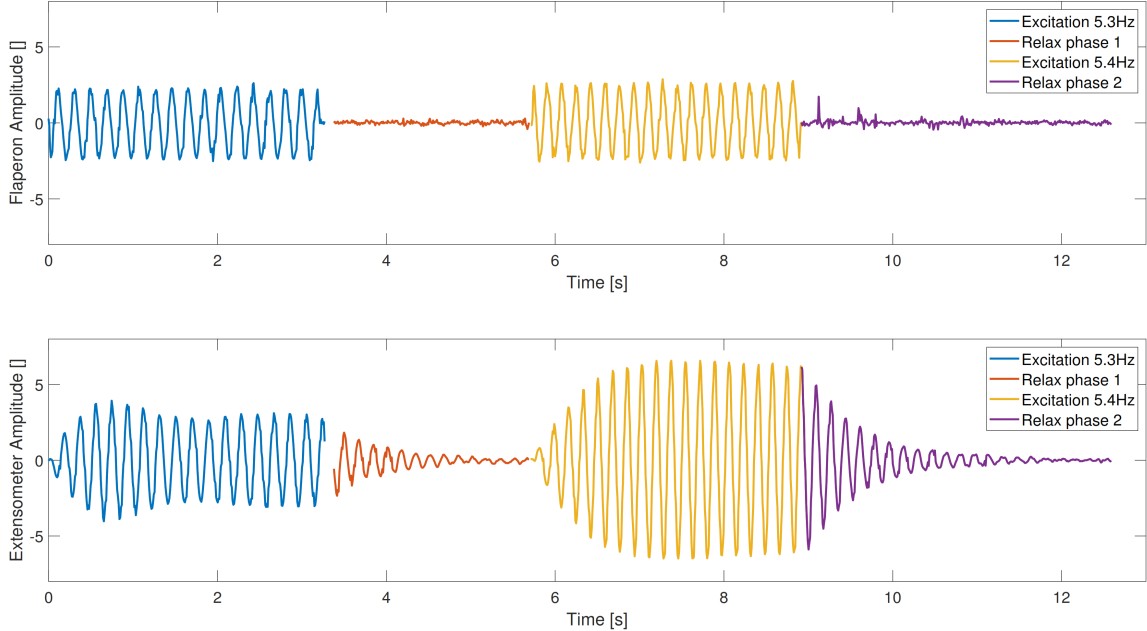

**Figure 4.** Sine Dwell sample plot extracted from real Flutter Flight Test data. The upper plot shows the forced excitation on the flaperon through the Flight Controls, while the lower plot shows the reading from the wing extensometers. In this sample only two excitation programs are depicted, while the full test point involves eight different excitation programs.

The paper will be divided into several sections. First the aircraft stores integration paradigm will be described in Section 2, supported by sample plots in Appendix A, which were extracted from custom software to analyze real flutter configurations. Then the impact of the phase angle and relationship to bandwidth on the identification of signals derived from Sine Dwell excitations will be demonstrated in Section 3. Next different techniques suitable for processing Sine Dwell signals will be described in Section 4, while a new robust data processing technique, the PRESTO algorithm, will be presented in

Section 5. In the experimental Section 6, the different techniques will be compared and verified over simulated data, focusing on the Laplace Wavelet estimation as the main source of comparison and further the PRESTO algorithm and Laplace Wavelet estimation will be validated with real Flutter Flight Test Data from envelope expansion Flight Tests, performed on Spanish Air Force F-18 during real stores integration campaigns. At last, Section 7 will extract several conclusions and future lines of work.

The scope of this paper is to study and compare different approaches for flutter data analysis on very short time series, where the ultimate goal is to accurately identify the different natural frequencies and damping factors of the aircraft at given dynamic pressures from real Flutter Flight Test data, assuming a quasi-stationary approach to the aeroelastic equations of motion described in the classical bibliography like [31–33]. The datasets analyzed will be limited to Sine Dwell excitations during the relax phase (after the excitation has ceased), considering very short time lengths of data, and in particular the present paper will be constrained to the Data Processing stage of the stores integration paradigm, which can be identified by the orange box in Figure 5. As a limitation of the experimental study, for classification reasons altitude, airspeed and stores configuration are not a disclosable piece of information from the real Flight Test data, so it will not be possible to present a full flutter analysis like the one in [34].

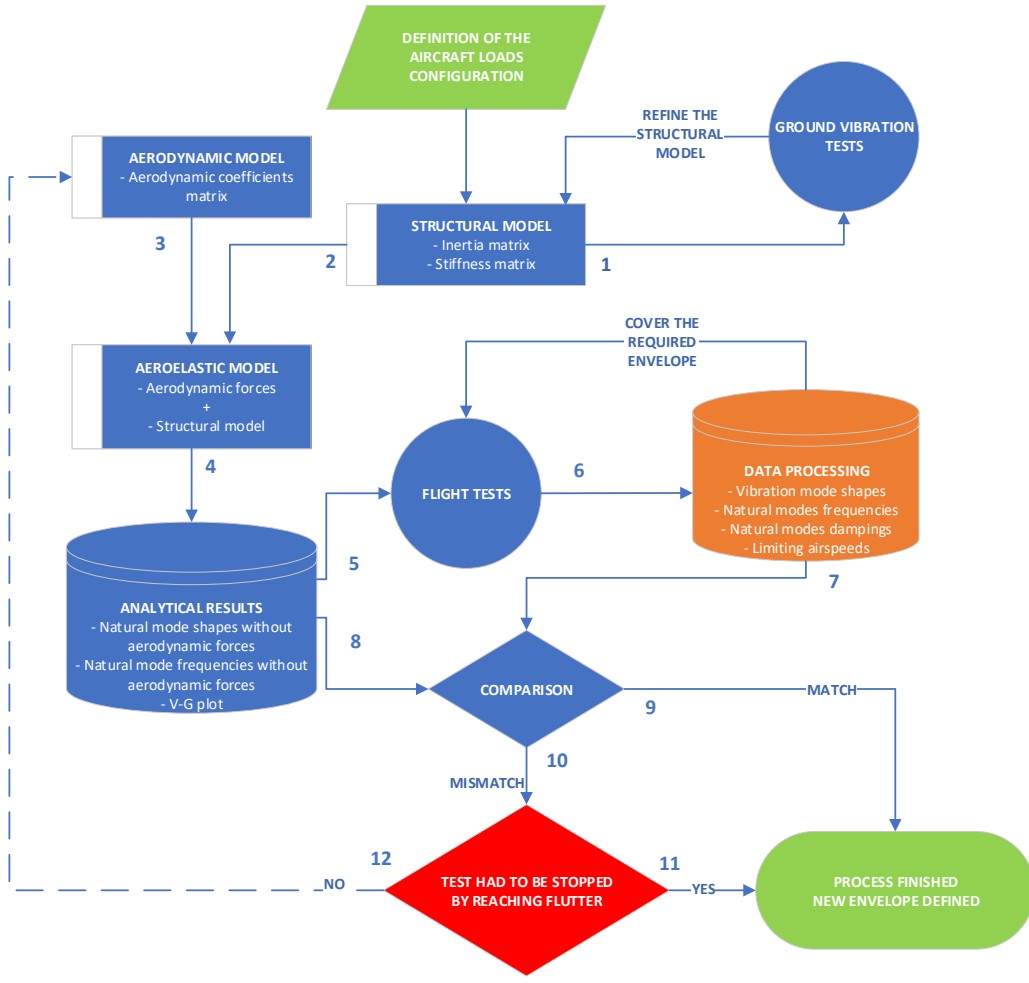

**Figure 5.** Stores integration paradigm particularized to Aeroelastic Flutter limiting factors. The stop point is reached either when flutter conditions have been found during flight or when the confidence on the accuracy of the model is high enough as to allow for a safe extrapolation of the flutter conditions.

## 2. Aircraft Aeroelastic Flutter and Stores Integration Paradigm

During Flutter Flight Tests the objective is to gather data aiming at characterizing the aeroelastic behavior of the aircraft and validate the aeroelastic simulation model. This is usually done both during and after each flight, and depending on the in-flight results, decisions will be taken to stop the test, start the analysis process again or proceed with more or less caution to the next Flight Test point. The envelope is eventually defined either because the flutter conditions have been reached during the flight (dangerous and hence undesirable) or because the confidence in the model is high enough as to allow for a safe extrapolation of the flutter point (hence the necessity to accurately characterize the real Flight Test data). Figure 5 describes the stores integration paradigm focused on Aeroelastic Flutter.

We will follow the paths described in Figure 2. The starting point is the definition of the stores configuration which requires clearing a new flight envelope. Take into account that all the subconfigurations need to be considered. For example, if two new fuel pods need to be integrated, it is necessary to consider as subconfigurations each partial fuel load of each pod and all the combinations between them. Then the parameters of all the subconfigurations are loaded into a Structural Model of the aircraft, where the natural modes of all the subconfigurations are calculated at zero airspeed (not considering aerodynamic forces). To validate this structural model we follow Path 1 and perform a Ground Vibration test (GVT) with the aircraft configured on the most significant subconfiguration. The results from the GVT will be used to refine the Structural Model. Once the Structural Model is considered accurate enough, Paths 2 and 3 feed the Aeroelastic model, where data from Path 2 come from the Aerodynamic Model, the known aerodynamic coefficients of the aircraft. Notice that those aerodynamic coefficients are usually defined for the aircraft baseline configuration and may not include the aerodynamic parameters of the structure considering those new stores. Therefore, the influence of the aerodynamics of the new stores is usually neglected. Although occasionally the influence is so big that the aeroelastic model is not reliable and the aerodynamic model requires some fine tuning, this situation is far from usual (Path 12). The Aeroelastic Model gives lead to Path 4 and returns the Analytical Results, which include the mode shapes from the Structural Model including natural frequencies (see Figure A2) at zero airspeed and the V-G plots (where "V-G" refer to velocity and damping factor, although frequencies are also plotted. See Figure A1). These results define the evolution of the natural frequencies and damping factors of the aircraft at different airspeeds along with the mode shapes for each subconfiguration, and therefore allow to select the most critical configurations to perform the flight tests, where the most critical configurations are selected considering the lowest airspeed where the damping factor of any mode becomes zero with an abrupt slope [7]. At the end of Path 4 the natural frequencies and modal shapes of the structure are obtained without the aerodynamic forces (zero airspeed). We want to highlight that calculating the mode shapes taking into account these aerodynamic forces for different points in the V-G plot would considerably improve the situational awareness of the test director during the test. However, in some cases such capability is not available, especially when the analysis is performed with proprietary software. Once the critical subconfiguration is selected, Path 5 is followed and flight tests are performed. The data gathered at the Flight Tests (Path 6) need to be processed, which is the actual scope of the present paper and repeated until the critical areas of the flight envelope are covered. After each flight, Path 7 and Path 8 compare the data between the real data results and the analytical data results. In case both results match and the confidence in the analytical model is considered sufficient, the pilot is cleared to proceed with the next test point and eventually finish the test with the envelope cleared (Path 9). In case of a mismatch there are two options. Either the flutter boundary has been reached, and hence the flutter boundary is known at that dynamic pressure (Path 11), or the flutter boundary has not been reached, but the mismatch differences call for a refinement on the analytic model, since proceeding would imply an unacceptable level of risk (Path 12). This last step requires to be followed with caution, since other factors

come into play. Defining a new aerodynamic model, including verification and validation, is extremely costly in time, resources (availability of wind tunnel and/or computational fluid dynamic models, software licenses, highly specialized personnel, etc.) and money. In the case that configuration is really necessary then it may be worth the effort. However, usually other solutions tend to be considered, like discarding the integration of that store for another of equivalent performance, limiting the envelope to safe known regions or selecting other subconfigurations where the aerodynamics of the store are not so relevant. This last situation is typical of wingtip missiles, where the aerodynamics tend to have a large impact. In those cases an option may be to discard the subconfigurations that include wingtip installation favoring underwing or fuselage positions, initially less prone to disturb the original aerodynamic model. However, there is not a fixed rule, and this case needs a thorough analysis that falls out of the scope of this paper.

## 3. Theoretical Characterization of the Frequency Response

The model of flutter equations in the classical bibliography assumes structural or equivalent viscous damping [29,33,35]. Assuming an underdamped system (damping factor $0 < \zeta < 1$) under quasi-stationary conditions, the aeroelastic equations of motion can be reduced to a homogeneous second order linear differential equation of constant coefficients. The derivation of such solution 1 can be found in multiple textbooks, like [36–38]:

$$x(t) = \sum_{j=1}^{K} a_j \cdot e^{-\zeta_j \omega_{nj} t} \sin\left(\omega_{dj} t + \phi_j\right) + \nu(t) \tag{1}$$

where $k$ is the number of different modes, $a$ is an amplitude constant, $\zeta$ is the damping factor, $\omega_n$ the natural angular frequency of the structure ($\omega_n = 2\pi f_n$, being $f_n$ the natural frequency), $t$ is time, $\omega_d = \omega_n \sqrt{1 - \zeta^2}$ is the damped angular frequency ($\omega_d = 2\pi f_d$ relates the damped angular frequency with the damped frequency $f_d$), $\phi$ is the phase angle and $\nu(t)$ represents structural and aerodynamic noise.

As was stated above, the objective of this paper is to propose a new technique to identify the frequency and damping factors of the different natural modes of the aircraft in flight. In order to do so, it is necessary to understand the behavior of the equations involved in the solution and also the different weight of each parameter. This theoretical characterization is important since it will aid us to better understand the problem of parameter estimation in the case of the presence of multiple modes. The focus will be set on the importance of the phase angle and the relationship between the bandwidth of the spectrum and the estimation of damping factors, following the Peak-Amplitude technique proposed by Ewins [29].

We will start considering the classical time series response for underdamped second degree linear ODE systems, with the constraint ($0 < \zeta < 1$), as indicated in Equation (1). For that particular case the Fourier transform of Equation (1) (multiplied by a step function to consider values from 0 to infinite) is:

$$X(\omega) = F\{x(t)\} = b \cdot \frac{(\alpha + i\omega)}{(\zeta\omega_n + i\omega)^2 + \omega_n^2(1 - \zeta^2)} = b \cdot \frac{(\alpha + i\omega)}{\omega_n^2 - \omega^2 + i2\zeta\omega_n\omega} \tag{2}$$

where:

$$\alpha = \frac{\omega_n \sqrt{1 - \zeta^2}}{\tan(\phi)} + \zeta\omega_n = \omega_n \left( \frac{\sqrt{1 - \zeta^2}}{\tan\phi} + \zeta \right) = \omega_n \beta \tag{3}$$

$$b = \frac{a}{\sqrt{2\pi}} \sin(\phi) \tag{4}$$

$$\beta = \frac{\sqrt{1 - \zeta^2}}{\tan\phi} + \zeta \tag{5}$$

Taking the absolute value of Equation (2) we obtain the power spectrum:

$$S_x(\omega) = |X(\omega)|^2 = b^2 \cdot \frac{(\omega^2 + \alpha^2)}{(\omega_n^2 - \omega^2)^2 + 4\zeta^2\omega_n^2\omega^2} = b^2 \cdot \frac{(\omega^2 + \alpha^2)}{\omega_n^4 + \omega^4 + \omega_n^2\omega^2(4\zeta^2 - 2)} \quad (6)$$

Note that the constant $b$ is a function of $\phi$. This expression is valid, except for the particular case with $\phi = k\pi$, $\forall k \in \mathbb{Z}$, which corresponds to the classical value of the damped sinusoidal. In this particular case the Fourier transform is:

$$X(\omega) = F\{x(t)\} = b' \cdot \frac{1}{(\zeta\omega_n + i\omega)^2 + \omega_n^2(1 - \zeta^2)} = b' \cdot \frac{1}{\omega_n^2 - \omega^2 + i2\zeta\omega_n\omega} \quad (7)$$

And the power spectrum:

$$S_x(\omega) = |X(\omega)|^2 = b'^2 \cdot \frac{1}{(\omega_n^2 - \omega^2)^2 + 4\zeta^2\omega_n^2\omega^2} = b'^2 \cdot \frac{1}{\omega_n^4 + \omega^4 + 2\omega_n^2\omega^2(4\zeta^2 - 2)} \quad (8)$$

where:

$$b' = b \cdot \omega_n \sqrt{1 - \zeta^2} \quad (9)$$

Please note here that we choose to group the amplitude terms $b$ and $b'$ with other parameters, since the proper amplitude of the response is not one of the parameters of interest for our estimations. The values of both $b$ and $b'$ will depend on the amplitude of the time response $a$, and since this value will also depend on the energy of the excitation and the gain of the sensors, the dependence of both $b$ and $b'$ with either the phase or the natural frequency cannot be used in real cases.

In this point, the present paper derives the equations assuming the time series have been windowed by a step function, since the general equations for a finite-time window function needlessly complicate the calculations. In experiments it is common practice to get a time sufficiently long so that the amplitude of the signal is below the noise level, so as a matter of fact the assumption of time infinitely long can be made without loss of generality, even though the data has actually been selected with a finite-time rectangular window. In general we can assume the scope of the present paper to be limited to spans of time $T$ where the data amplitude of the time series at $t = T$ is similar to the level of noise.

The decision on not to window the results with a common window (hanning, blackman, etc.) is deliberate. Coll [18] employs a Hann window in her JFlutter software, but her data come from Frequency Sweeps 30 s to 60 s long. On the other hand, Potts [20] suggests that windowing is not necessary to obtain accurate data with data lengths and frequencies similar to the present case. Considering that the aim of this paper is to support the results of aeroelastic data on aircraft structures, with frequencies between 2 Hz and 10 Hz and time samples of less than 5 s, searching for exponentially damped sinusoidals arising from Sine Dwell excitations, given the low amount of cycles in the data, the disturbance in amplitudes from a window function, different from a boxcar window, would throw a significant error on top of the error inherent to the processing methods with such low amount of data. This particular statement has been confirmed by the authors in preliminary experiments.

In any standard spectral analysis two factors are typically considered: the resonance frequency and the bandwidth. In this point of the theoretical analysis we will focus on deriving these two values analytically, with the objective of determining their relationship with the parameters of the signal ($\omega_n$, $\zeta$ and $\phi$).

### 3.1. Resonance Frequency of a Single Mode

Once calculated the power spectrum of a single mode (Equations (6) and (8)), in order to characterize the resonance frequency we must calculate the first derivative in search for the points of maxima. Now let us determine the maximum of the power spectrum for a single mode as related to $\omega_n$, which will be the resonance angular frequency $\omega_0$. For this purpose we will partially derive Equation (6) with respect to $\omega$ and set it equal to zero.

Equation (8) is a particular case of the former when the phase angle $\phi = k\pi$ and will be treated accordingly when analyzing the derivative.

$$\left.\frac{\partial S_x(\omega)}{\partial \omega}\right|_{\omega=\omega_0} = b^2 \left[\frac{2\omega_0(\omega_n^4 + \omega_0^4 + \omega_n^2\omega_0^2(4\zeta^2 - 2))}{(\omega_n^4 + \omega_0^4 + \omega_n^2\omega_0^2(4\zeta^2 - 2))^2} - \frac{(\omega_0^2 + \alpha^2)(4\omega_0^3 + 2\omega_0\omega_n^2(4\zeta^2 - 2))}{(\omega_n^4 + \omega_0^4 + \omega_n^2\omega_0^2(4\zeta^2 - 2))^2}\right] = 0 \qquad (10)$$

So, removing the denominator and simplifying we get Equation (11).

$$(\omega_n^4 + \omega_0^4 + \omega_n^2\omega_0^2(4\zeta^2 - 2)) - (\omega_0^2 + \alpha^2)(2\omega_0^2 + \omega_n^2(4\zeta^2 - 2)) = 0 \qquad (11)$$

Notice that the term $\omega_0$ was simplified accounting for the trivial solution $\omega_0 = 0$. However, calculating the second derivative it can be demonstrated that this particular result is a minimum. The demonstration is out of the scope of this paper.

Rearranging the terms and simplifying Equation (11) in order to solve $\omega_0$:

$$\omega_0^4 + 2\alpha^2\omega_0^2 + \alpha^2\omega_n^2(4\zeta^2 - 2) - \omega_n^4 = 0 \qquad (12)$$

This equation can be solved (please take into account that we only allow positive real values for $\omega_0$), obtaining the resonance angular frequency with Equation (13)

$$\omega_0 = \sqrt{-\alpha^2 + \sqrt{\alpha^4 - 4\zeta^2\alpha^2\omega_n^2 + 2\omega_n^2\alpha^2 + \omega_n^4}} \qquad (13)$$

Now, replacing $\alpha$ with Equation (3).

$$\omega_0 = \omega_n\sqrt{-\beta^2 + \sqrt{\beta^4 + 2(1 - 2\zeta^2)\,\beta^2 + 1}} = \omega_n\,\gamma \qquad (14)$$

Please note here that $\beta$ (see Equation (5)) depends on both the damping factor and the phase of the original signal. So, although the resonance frequency is proportional to the natural frequency of the damped signal, it will also depend on both the phase and the damping factor. This dependency makes it difficult to directly estimate the natural frequency from the resonance frequency, as one might originally expect. To better understand the degree of dependency of $\gamma$ with $\zeta$ and $\phi$ we will now study its range of variation.

In general, for underdamped signals we have that $0 \leq \zeta \leq 1$. Then $0 \leq \zeta^2 \leq 1$ and $1 \geq 1 - 2\zeta^2 \geq -1$. So, considering $1 - 2\zeta^2 \leq 1$ we can deduce that $\beta^4 + 2(1 - 2\zeta^2)\beta^2 + 1 \leq \beta^4 + 2\beta^2 + 1$. Taking this into consideration we can establish the upper limit to the value of $\omega_0$:

$$\omega_0 \leq \omega_n\sqrt{-\beta^2 + \sqrt{\beta^4 + 2\,\beta^2 + 1}} = \omega_n \qquad (15)$$

In this inequation, boundary solution $\omega_0 = \omega_n$ is achieved when $\beta = 0$. So, considering the definition of $\beta$ given by Equation (5), this particular solution will be achieved in the case of $\phi = \phi_1 = -\arctan\sqrt{1 - \zeta^2}/\zeta$. Please notice that this is true also when $\zeta = 0$ for any value of $\phi$.

Now let us look for the lower limit of $\omega_0$. We consider that $(1 - 2\zeta^2)^2 \leq 1$, and thus we can establish that $\beta^4 + 2(1 - 2\zeta^2)\beta^2 + 1 \geq \beta^4 + 2(1 - 2\zeta^2)\beta^2 + (1 - 2\zeta^2)^2$. So, taking again into consideration Equation (14) and simplifying we can establish that:

$$\omega_0 \geq \omega_n\sqrt{-\beta^2 + \sqrt{\beta^4 + 2(1 - 2\zeta^2)\,\beta^2 + (1 - 2\zeta^2)^2}} = \omega_n\sqrt{1 - 2\zeta^2} \qquad (16)$$

As a conclusion of this demonstration, in general for underdamped signals the resonance angular frequency $\omega_0 \geq \omega_n\sqrt{1 - 2\zeta^2}$. That is, independently on the phase, the frequency of the maximum of the power spectrum will be greater than $\omega_n\sqrt{1 - 2\zeta^2}$. This limit case will be achieved for the particular case of $\phi = \phi_2 = k\pi$, $\forall k \in \mathbb{Z}$ which can be deduced from the maximization of Equation (8). In that particular case maximizing the

power spectrum implies the minimization of the denominator, which after simplifying leads to $\omega_0^2 + \omega_n^2(1 - 2\zeta^2) = 0$. From this point it is easy to see that effectively the value $\omega_0 = \omega_n\sqrt{1 - 2\zeta^2}$ is the positive solution of this equation.

From these upper and lower limits of $\omega_0$ we can deduce that the estimation of the natural angular frequency $\omega_n$ from the amplitude of the power spectrum is not straightforward as some might expect, since the value of the resonance frequency will also depend on the damping factor and on the phase of the signal.

To better understand these expressions, Figure 6 shows the relationship between the relative resonant frequency ($\gamma = \omega_0/\omega_n$) of a single underdamped mode and the damping factor ($\zeta$) for different phase values ($\phi$). As we can see, the two extreme values obtained with $\phi_1$ and $\phi_2$ represent both extremes of the curves. It is important to note the strong relationship between the phase angle and the resonance frequency in the power spectrum. This statement will be relevant in the upcoming sections.

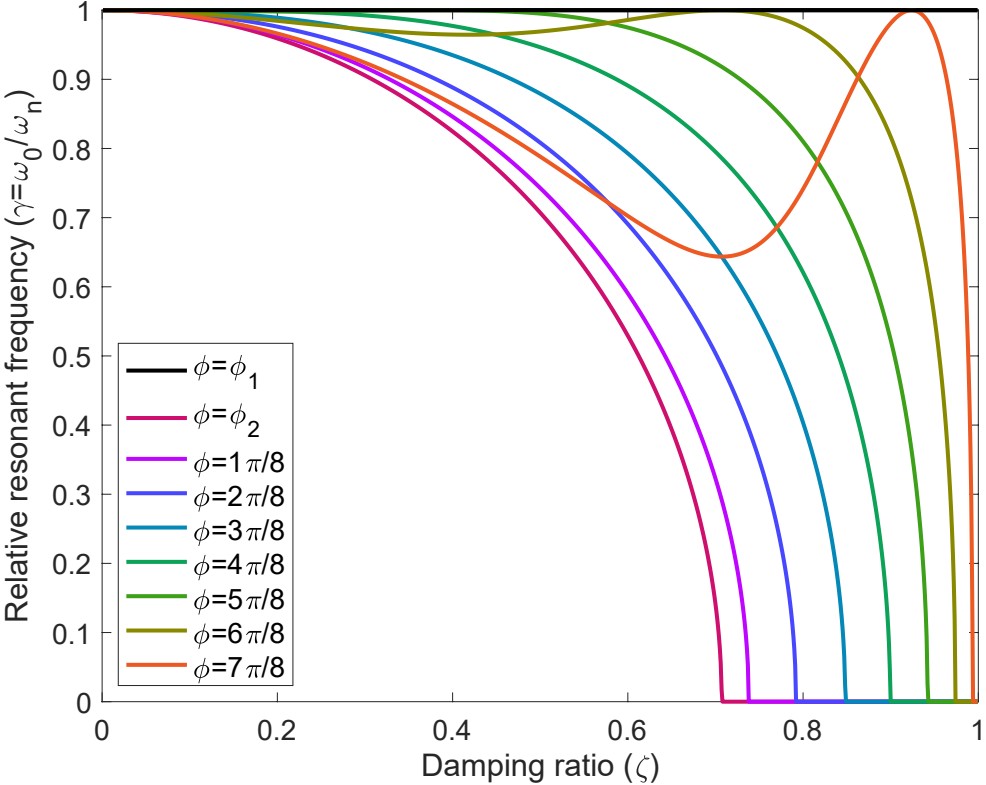

**Figure 6.** Relative resonant frequency ($\gamma = \omega_0/\omega_n$) of a single underdamped mode, with respect to the damping factor ($\zeta$) for different phase values ($\phi$).

### 3.2. Bandwidth of a Single Mode

Another factor to assess the impact of the different parameters when identifying the system is the bandwidth. Ewins in [29] describes the Peak-Amplitude method to estimate damping factors. However, this approach leaves behind the phase angle, which depending on the relationship with the rest of the parameters may introduce a large source of error.

To analytically study the bandwidth first we need to determine the maximum amplitude of the power spectrum at the resonance angular frequency $\omega_0$, which will be obtained by evaluating Equation (6) in $\omega = \omega_0 = \omega_n\gamma$ as described in Equation (14).

After simplifying, we can determine $\max\{|X(\omega)|^2\} = |X(\omega_0)|^2$ using Equation (17).

$$|X(\omega_0)|^2 = b^2 \cdot \frac{\gamma^2 + \beta^2}{\omega_n^2(1 - \gamma^2)^2 + 4\zeta^2\omega_n^2\gamma^2} \tag{17}$$

Now to determine the 3 dB bandwidth we must look for the frequency values $\omega_1$ and $\omega_2$ that make $|X(\omega)|^2/|X(\omega_0)|^2 = 1/2$.

$$\frac{|X(\omega)|^2}{|X(\omega_0)|^2} = \frac{\frac{\omega^2 + \omega_n^2 \beta^2}{(\omega_n^2 - \omega^2)^2 + 4\zeta^2 \omega_n^2 \omega^2}}{\frac{\gamma^2 + \beta^2}{\omega_n^2 (1 - \gamma^2)^2 + 4\zeta^2 \omega_n^2 \gamma^2}} = \frac{1}{2} \tag{18}$$

Rearranging the terms and simplifying we can get an equation $\omega^4/2 + g\omega_n^2\omega^2 + h\omega_n^4/2 = 0$ that can be solved, being $g$ and $h$ given by Equations (19) and (20), respectively.

$$g = 2\zeta^2 - 1 - \frac{(1 - \gamma^2)^2 + 4\zeta^2\gamma^2}{\gamma^2 + \beta^2} \tag{19}$$

$$h = 1 - \beta^2 \frac{2(1 - \gamma^2)^2 + 8\zeta^2\gamma^2}{\gamma^2 + \beta^2} \tag{20}$$

At last, considering that $\omega > 0$, the two valid solutions of the equations $\omega_1$ and $\omega_2$ will be given by Equation (21).

$$\omega_{1,2} = \omega_n \sqrt{-g \pm \sqrt{g^2 - h}} \tag{21}$$

Thus the 3 dB bandwidth ($B$) will be determined using Equation (22).

$$B = \omega_1 - \omega_2 = \omega_n \left( \sqrt{-g + \sqrt{g^2 - h}} - \sqrt{-g - \sqrt{g^2 - h}} \right) \tag{22}$$

Which after some simplification yields:

$$B = \omega_n \sqrt{2} \sqrt{-g - \sqrt{h}} \tag{23}$$

As we can see, the bandwidth will be proportional to the natural frequency, but again it strongly depends on the damping factor and the phase of the signal. To better understand this dependency, let us analyze the bandwidth for the two extreme phase cases $\phi_1$ and $\phi_2$ considered above.

As was already stated, the particular case of $\phi = \phi_1$ was associated to $\beta = 0$, $\gamma = 1$ and $\omega_0 = \omega_n$, which was one of the extreme values of the resonance angular frequency. Replacing those values in Equations (19) and (20) we get $g = 2\zeta^2 - 1 - 4\zeta^2 = -1 - 2\zeta^2$, and $h = 1$. So, substituting $g$ and $h$ in Equation (23) we get (24).

$$B|_{\phi_1} = \omega_n \sqrt{2} \sqrt{1 + 2\zeta^2 - \sqrt{1}} = 2\zeta\omega_n \tag{24}$$

On the other hand, the other extreme case was obtained when $\phi = \phi_2$, for which $\omega_0 = \omega_n \sqrt{(1 - 2\zeta^2)}$. In this case the maximum value of the power spectrum must be determined substituting this condition in Equation (8) and simplifying.

$$|X(\omega_0)|^2 = b^2 \cdot \frac{1}{4\omega_n^4 \zeta^2 (1 - \zeta^2)} \tag{25}$$

Thus, in the case of $\phi = \phi_2$ to determine the bandwidth we must solve Equation (26).

$$\frac{|X(\omega)|^2}{|X(\omega_0)|^2} = \frac{4\omega_n^4 \zeta^2 (1 - \zeta^2)}{(\omega_n^2 - \omega^2)^2 + 4\zeta^2 \omega_n^2 \omega^2} = \frac{1}{2} \tag{26}$$

Simplifying we obtain the same equation $\omega^4/2 + g\omega_n^2\omega^2 + h\omega_n^4/2 = 0$ but in this case with $g = 2\zeta^2 - 1$ and $h = 8\zeta^4 - 8\zeta^2 + 1$. Considering these values of $g$ and $h$ in Equation (23) we obtain the bandwidth for the extreme case with $\phi = \phi_2$ in Equation (27).

$$B|_{\phi_2} = \omega_n \sqrt{2} \sqrt{1 - 2\zeta^2 - \sqrt{8\zeta^4 - 8\zeta^2 + 1}} \tag{27}$$

To better understand these expressions, Figure 7 shows the relationship between the relative bandwidth ($B/\omega_n$) of a single underdamped mode, and the damping factor $\zeta$ for different phase values. As we can see, the two extreme values obtained with $\phi_1$ and $\phi_2$ represent both extremes of the curves, but in this case the conditions are different for low damping and high damping factors. For low damping factors, the bandwidth is practically independent of the phase angle. However, for relatively high values of the damping factors (but always within our values of interest), there is significant disturbance to the linearity of the relationship between the damping factor and the phase angle.

This result is interesting if the Peak-Amplitude technique [29] is to be employed, where it is necessary to remember that unless the conditions favor closeness between the resonance frequency and the natural frequency, or low damping factor values are considered, phase must be taken into account to get an accurate damping factor estimation. In addition, take into consideration that in the case that several modes are involved, the Peak-Amplitude technique would not throw any useful result.

Regarding the existence of one single or several natural modes, it is important to notice that the analysis performed in this section has taken into account signals with one single mode. The theoretical analysis has shown the strong relationship between the phase angle $\phi$, the natural angular frequency $\omega_n$ and the damping factor $\zeta$ related to the power spectrum. This result can be extrapolated qualitatively to more than one natural mode although not quantitatively. However, this approximation will be sufficient to calculate the seeds for the PRESTO algorithm, as will be described in Section 4.4.

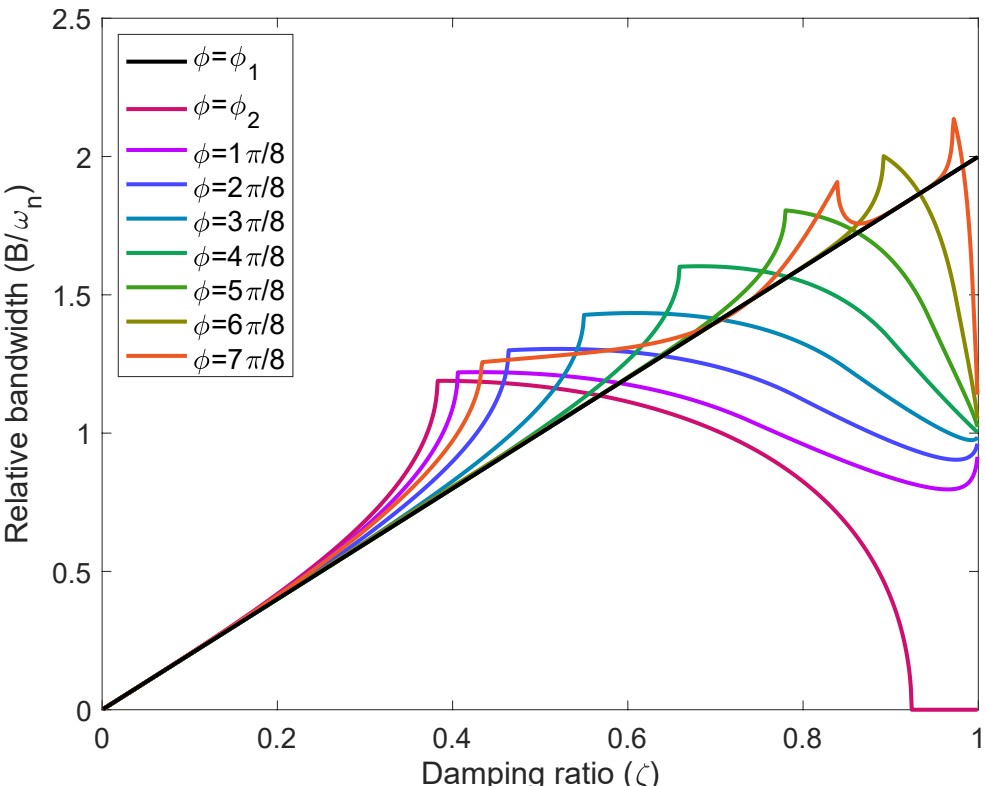

**Figure 7.** Relative bandwidth ($B/\omega_n$) of a single underdamped mode, with respect to the damping factor ($\zeta$) for different phase values ($\phi$).

## 4. Data Processing Paradigm, Optimization Algorithms and Fitting Functions

Once the theoretical framework has demonstrated the importance of the phase in the estimation of both the natural frequency and the damping factor, this section will describe different optimization techniques employed to characterize signals similar to Sine Dwells in terms of duration and frequency range.

In the introduction 1 several techniques were described to process aeroelastic sets of data coming from Flutter Flight Tests. However, only a few were suitable for Sine Dwell excitations. Please notice one very important point. The main constraint on these techniques is the final form solution defined by the certification standards for airworthiness (mainly [6,7]). Those standards require a very specific numerical limit on damping factor of the critical mode, and therefore it is necessary to get the best reconstruction possible with the closest functions to the critical natural modes not simply an accurate reconstruction of the signal achieved by a different model or as a sum of different modes. For example, a classical ARMA model might produce an accurate reconstruction of the signal. However, the parameters of the model do not fit the requirements of the mentioned standards, which can only be fulfilled by a model based on Equation (1).

### 4.1. Matrix Pencil Algorithm

Matrix Pencil is a variant of the Prony Method, and a description of the technique can be found in [19–22]. It is based on the same principle as the Prony Method, a decomposition of the signal in sums of exponentially damped sinusoids, with the particularity that instead of decomposing in as many modes as possible considering the number of samples in the signal (as for the Prony Method), it is possible to select a number of signals so that the fit is the best possible with that number of elements.

This technique is extremely accurate and useful to reconstruct Sine Dwell signals, since implicitly filters the lowest energy modes with SVD (Singular Value Decomposition) and the reconstruction is very accurate and fast. However, the signal model requires a high order to reconstruct the signal accurately, and when comparing all those modes to the actual data during the simulation runs, the error returned is considerably high compared to other techniques.

### 4.2. Wavelet Techniques

Different wavelet techniques have been developed on this topic. The wavelet transform seems the perfect candidate to deal with such signals, short length and low frequency, and in particular the Laplace Wavelet transform has returned excellent results on flutter Sine Dwell signals.

The Laplace Wavelet has been successfully employed by Freudinger, Lind and Brenner [12,13] on this same problem and aircraft. There are different ways to fit the sampled signal to the estimation, and in particular one way employed by these authors was the Matching Pursuit algorithm [39,40]. A broader discussion on this technique will follow in Section 6.2.3.

### 4.3. Peak-Amplitude Technique

This technique was described by Ewins in Reference [29]. Although the technique is too basic and the results are not accurate when several modes come into play, due to its extremely fast estimation, the authors will follow this approach to estimate the frequency and damping factor seeds in the PRESTO technique but updated by employing the equations developed in Section 3.2. More information on this aspect will be provided in Section 5.

### 4.4. Proposed Paradigm

Considering the drawbacks from the aforementioned techniques, the authors decided to follow a different approach and paradigm described in Figure 8, which shows the standard optimization process used to determine the parameters of the signal. Please notice that this paradigm, although common, is not necessarily followed by the techniques described above. It is also important to remark that the current paper is based on a developmental procedure oriented to comparing different techniques. The present paper is extracted from an early -yet promising- stage, and hence the main two metrics employed are accuracy (error between truth and estimation) and processing time. To prevent bias in

the processing time, the stop criterium selected was to complete the selected number of iterations. In particular, 10 runs and 1000 iterations each run were performed.

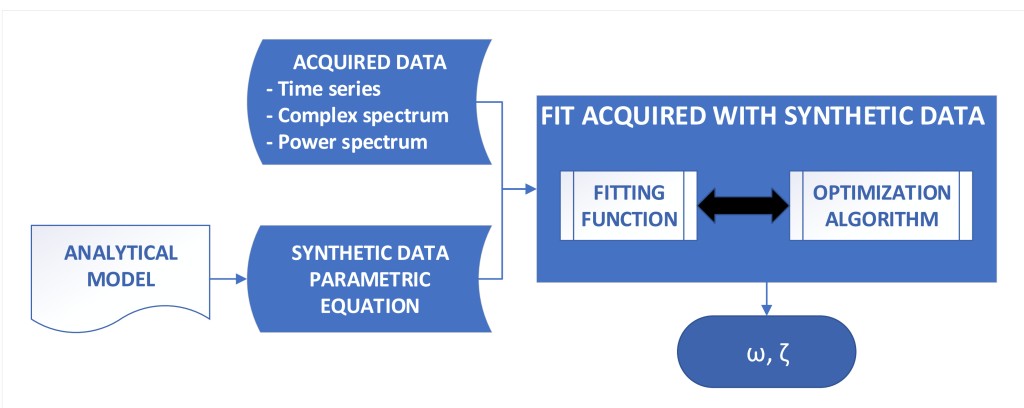

**Figure 8.** Data processing paradigm employed.

We will now focus on two mode responses. In real scenarios it is quite common to find close modes which overlap both in time and frequency domains. Thus we will consider Equation (28) as the main analytical model used to describe the basic parametric equation to be used by the fitting algorithm.

$$\hat{x}(t) = a_1 \cdot e^{-\zeta_1 \omega_{n1} t} \sin (\omega_{d1} t + \phi_1) + a_2 \cdot e^{-\zeta_2 \omega_{n2} t} \sin (\omega_{d2} t + \phi_2) + \nu(t) = x(t) + \nu(t) \quad (28)$$

In this case $\hat{x}(t)$ represents the signal measured from the acquired data, $x(t)$ the signal reconstruction and $\nu(t)$ represents higher order terms, including noise. The first thing to consider in the optimization process is the domain in which optimization of the fitting function is defined. That is, we can define the mean square error either in the time domain, in the complex frequency domain or in the power spectrum. All those cases will be analyzed below.

- *Time domain fitness function*. This is the most basic approximation. The basic parametric Equation (28) feeds the optimization algorithm, where it generates a synthetic time series. This synthetic time series, along with the sampled time series, feeds the fitting function and the cycle is repeated until the desired stop criterion is reached. So, in the time domain, the mean square error is defined as $MSE_{time} = E\{(x(t) - \hat{x}(t))^2\}$. The main disadvantage of this fitness function is that random seeds sometimes converge to local minima, so it is necessary to find an accurate estimator.

- *Complex frequency domain fitness function*. As an alternative approach one standard possibility is the use of a fitness function defined in the frequency domain. In this case, the parametric function passed to the optimization algorithm is also Equation (28). However, after reconstructing the synthetic time series it numerically calculates the complex spectrum and separates the real and imaginary parts. On the other hand, the sampled time series is transformed numerically into a frequency spectrum, also separating real and imaginary parts. The synthetic and sampled data are fed to the fitting function and the cycle is repeated until the stop criterion is reached. So, in this case the fitness function is defined as $MSE_{freq} = E\{|X(\omega) - \hat{X}(\omega)|^2\}$, where $\hat{X}(\omega)$ is the Fourier transform of the measured signal $\hat{x}(t)$ in the form stated above.

With this method convergence problems and local minima are still present. Although the use of frequency responses in which the energy of the signal is concentrated in less samples allows for the reduction in the amount of frequency terms, reducing the computational cost of the evaluation of the fitness function, the more complex process of extracting the Fourier transform of the signal compensates for the increased speed.

- *Power spectrum based fitness function*. The power spectrum approximation follows the same approach as the complex frequency approximation, but in this case the power spectrum is numerically calculated from the complex frequency, instead of having both real and imaginary parts separated. The fitting function to be optimized is evaluated from the power spectrum of the signals, $MSE_{spec} = E\{(|X(\omega)|^2 - |\hat{X}(\omega)|^2)^2\}$. This choice tends to reduce the convergence problems, but the solutions achieved sometimes imply a loss in performance with respect to the two first fitness functions. In this case, as in the *Complex Frequency domain estimation*, it is possible to reduce the amount of frequency terms to be taken into consideration in order to reduce the computational cost of evaluating the fitness function without local minima problems, but the convergence tends to be very slow and requires multiple (usually more than six) runs to converge.

Taking into consideration the main advantages and disadvantages of each fitness function, the next sections proposes a novel approach that tries to combine the benefits of time domain optimization while reducing the convergence problems of the optimization algorithm.

## 5. Proposed Combined Power Spectrum and Time Optimization

This section describes the proposed estimation approach, denoted the combined PoweR spEctrum Short Time Optimization (PRESTO). Analyzing the conclusions derived from the fitting functions described in the previous section, we can establish several facts:

- The Time and Complex frequency domain fitness functions are very sensitive to local minima;
- The initial conditions make a difference in terms of convergence to local or global minimum;
- The power spectrum fitness function can reach the global minimum but after many iterations and is insensitive to initial conditions;
- The Time and Complex Frequency fitness functions quickly reach the minimum but are very sensitive to initial conditions and prone to reach local minima if the seeds are not adequate.

With these facts in mind the objective of the proposed method is to combine a good estimation of the seeds with the Peak-Amplitude estimation and employ the time based estimation to return the final solution. The detailed process is as follows:

1. Estimate the natural frequency seed as the peak frequency in the power spectrum;
2. Estimate the damping factor, with the 3 dB bandwidth and the natural frequency known, solving from Equation (24). Notice that the usual range of damping factors in real applications is not very different from this estimation;
3. Since the phase term $\phi$ and amplitude $a$ are unknown at this point, they are randomly initialized;
4. Run the Time based estimation with the seeds obtained from the steps above.

The proposed estimation approach is sensibly faster than other methods, both classical and proposed by the paradigm, and the experimental results on synthetic data are really promising.

## 6. Experimental Results

The objective of this section is to describe the experimental setup and to analyze the results in order to evaluate the proposed estimation technique and in order to further verify the conclusions extracted from the theoretical analysis carried out in the previous sections. For this purpose, two groups of experiments are included:

- First set of experiments processing synthetic data:
  - The primary objective is to measure the accuracy of the described methodologies, evaluating the errors in frequency and damping factors between the estimations and the ones used to generate the signals;

-   The secondary objective is to measure the computational cost of the different techniques.

-   Second set of experiments processing real Flight Test Data, obtained from Flutter Flight Tests performed on Spanish Air Force F-18:

    -   The best two techniques obtained from the synthetic data experiments will be employed;
    -   The error will be assessed by performing a linear regression between the measured data and the reconstructed signal with the processed parameters and comparing the MSE of the normalized signals (normalized against the maximum amplitude of each real signal).

### 6.1. Multimodal Estimation of Simulated Data

6.1.1. Description of the Experiment Conditions

As a combination of two different modes following Equation (28), 10.000 simulated data signals have been generated considering these constraints:

-   Each signal lasts 5 s;
-   The sampling frequency is 85 Hz;
-   The parameters of each mode and signal were randomly selected from this range of parameters (notice the number of significant digits):

    -   Natural frequency: $3.0 \text{ Hz} \leq f_n \leq 6.0 \text{ Hz}$;
    -   Damping factor: $0.03 \leq \zeta \leq 0.20$;
    -   Phase angle: $0.00 \leq \phi \leq 2 \cdot \pi$;
    -   Amplitude: $0.01 \leq a \leq 0.50$;
    -   White Gaussian noise SNR: 0 dB, 5 dB, 10 dB.

The optimization algorithm considered by the authors for the paradigm described in Figure 8 is a standard "Trust Region" algorithm [41] and the error metric Mean Square Error (MSE). The boundaries employed to randomly initialize each parameter are broad compared to the expected real (and simulated) values. In the case of the phase angle $\phi$ the whole $2\pi$ circumference, while the others varied from $[-10\ 10]$ for the amplitude constant $a$, $[0\ 1]$ for the damping factor $\zeta$ and $[0\ 10]Hz$ for the natural frequency $f_n$. As we can see, the parameter margins for the simulated data were representative of real cases. The selection of the optimization algorithm was not random and several options were considered in the preliminar experiments, including the Levenberg–Marquardt method, Genetic Algorithms and the Trust Region algorithm. After these preliminary experiments and several considerations, the selected one was the Trust Region, since it tends to perform reasonably fast, it is robust, and it can be applied to ill-conditioned problems [42].

In order to minimize the convergence towards local minima 10 runs of each algorithm were executed, with a maximum of 1000 iterations on each run. The stop criterium selected was to stop after all the runs (and iterations) were performed, to account for a uniform time estimation. In the case of the PRESTO technique, the seeds were obtained from the Peak-Amplitude estimation with Equation (24).

In the complex frequency based and the power spectrum based fitting functions we removed frequency values far from the values of the signals expectations. In particular, frequencies higher than 14 Hz were discarded.

Regarding Matrix Pencil, Peak-Amplitude and Laplace Wavelet techniques, given the deterministic characteristics of the techniques it was not necessary to execute 10 runs. Instead 1 single run was executed, and hence the plots show one straight line at a constant error level.

Matrix Pencil estimation was performed considering order 200 and an order reduction of 196 natural modes from those original 200 (keeping 4 natural modes for the final estimation). These parameters are similar to the ones employed in the referenced bibliography. One important point to remark is that those parameters were selected by the authors after trying different combinations and selected the ones with lower error. It is necessary to

mention that the process was intensively tailored by the authors attempting to improve the results with different combinations of parameters. The details of the technique including the order reduction can be found in [19–21]. The technique returned 4 different natural frequencies and their corresponding damping factors. In order to select the best matches, a first attempt was made following [43], where an energy sorting technique is employed. However, the best matching frequencies returned were completely different to the expected ones. At last, the modes were sorted by frequency and the closest match to the expected frequencies was selected.

Laplace Wavelet estimation was made following [13] in a Matching Pursuit scheme. The technique can be run multiple times to return any number of modes, so the authors applied the technique expecting two different modes. Regarding the dictionaries, the ranges were substantially different from the ones used by the authors in the other techniques, since the size of the dictionaries was too large to achieve results in a reasonable time; therefore, the dictionary range was limited to the range of parameters used to generate the signal and the number of significant digits the minimum used for each parameter. Regarding the time delay, the range was $[-5.0 \ +5.0]$ seconds. From this point of view, the Laplace Wavelet estimation had better conditions than the PRESTO algorithm.

Regarding the error metrics, since the samples were generated artificially it is possible to determine the accuracy in the estimation of the relevant parameters. For this purpose, the authors measured the average relative error (%) in the estimation of the natural frequency $\omega_n$ and the Root Mean Square Error (RMSE) in the estimation of the damping factor $\zeta$.

### 6.1.2. Analysis of Results

Figures 9 and 10 show the behavior of the estimation techniques for the natural frequency and the damping factor respectively, and Table 1 includes the numerical results of the estimations considering time consumed and different signal to noise ratios (SNRs).

The time based estimation and the complex frequency based estimation obtain precise estimations with 6 repetitions of the optimization process. However, they are very prone to get stuck in local minima. On the other hand, the power spectrum based estimation tends to find the global minimum, but at the cost of requiring 10 repetitions (and consequently time) to obtain good results. These three techniques show poor performance regarding time consumed, although the error estimations show reasonably good results among the analyzed techniques. It is worth noticing that the techniques are sensitive to noise, showing a great difference between the 0 dB SNR datasets' solutions (similar to the Peak-Amplitude estimations) and the 10 dB datasets' solutions (similar to the PRESTO estimation).

The Peak-Amplitude estimation is the simplest of all and shows a reasonably good estimation considering the time required, several orders of magnitude faster than all the others. The technique is rather insensitive to noise, although it is expected that outliers will affect it enormously, especially for the frequency estimation (and hence to the damping factor estimation).

Matrix Pencil estimation returns bad results related to both frequency and damping factor estimations, although the time required scored a very low value and is extremely insensitive to noise. The technique requires numerous signals to return good results to reconstruct the original signal. The authors performed numerous tests on the datasets, exploring heuristically different modes considered (accuracy of the model) and reduction orders (noise filtering), and the best results related to the metrics required by our problem included 200 modes and filtering 196 signals (keeping the 4 different signals with most energy to reconstruct the original signal).

The Laplace Wavelet technique is also, along with the Matrix Pencil estimation, extremely insensitive to noise. Even though under a high SNR the results are below the average, notice that under noisy conditions the results are considerably good, and in fact the best of all the techniques for $SNR = 0$ in damping factor estimation. From the computational load aspect, the technique returned a well below average time result (lowest

equals best), and the time required for each estimation is good enough to process sensors in real time (maybe a couple seconds delay, depending on the dictionary size).

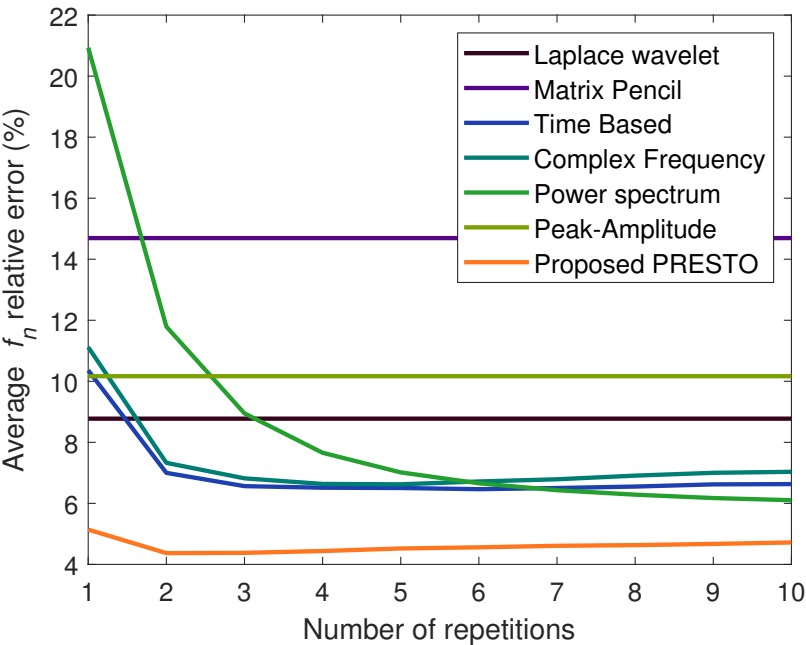

**Figure 9.** Average relative error (%) in the estimation of the natural frequency ($f_n = \omega_n/(2\pi)$) of two simulated underdamped modes. SNR 5 dB.

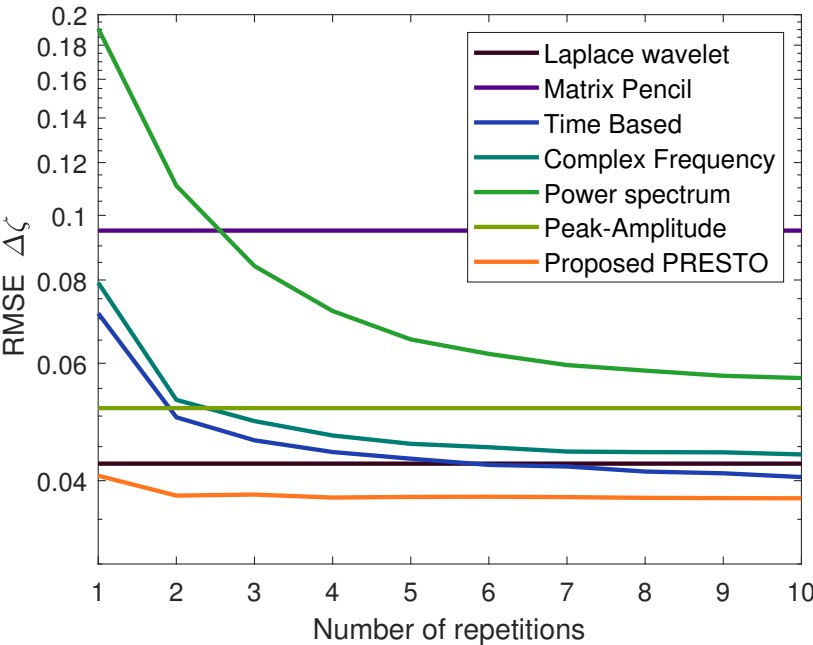

**Figure 10.** Root mean square error (RMSE) in the estimation of the damping factor ($\zeta$) of two simulated underdamped modes. SNR 5 dB. Logarithmic scale

The proposed PRESTO technique, described in Section 5, starts calculating the Peak-Amplitude estimation to provide seeds for the Time based estimation. The results show a much lower error for the proposed solution, both in the estimation of the natural frequency and the damping factor, than any of the other techniques alone. The technique is very sensitive to noise. However, it is important to remark that, even with the worst noise

conditions considered ($SNR = 0$), the technique achieved the best results in frequency and second best in damping factor estimations. Timewise, the technique matches in time the Laplace Wavelet technique and also poses as a candidate for real time estimations. It is also interesting to note that the Time based estimation itself, seeded with random values, did not initially show spectacular behavior. However, combined with an extremely fast but low accuracy estimation, it turned the tides and resulted in a very accurate technique in most of the cases.

To better analyze the problems of convergence of the different solutions, Figures 11 and 12 show the results after removing the 10% worst estimations. Notice that many of those datasets include the lowest frequency and highest damping factor combinations resulting in very short datasets, occasionally not even completing one full cycle before the signal is completely dampened. Now the first two approaches (time based estimation and complex frequency estimations) show a much better result when compared to the power spectrum based technique and eventually lead to results similar to the PRESTO estimation. Regarding the Laplace Wavelet estimation, the results show a slightly improved behavior than with the full dataset considered, meaning that even with short timed and heavily damped signals the technique is mostly unaffected, a fact that plays in its favor. The PRESTO technique, however, sensibly improved the estimation meaning that, as expected, short timed signals heavily impact the accuracy of the estimations.

In order to analyze the computational cost and the dependency of the results with the SNRs, numerical results are detailed in Table 1 with different SNRs (0 dB, 5 dB and 10 dB). The plots Figures 9–12 show the relationship between different estimators considering only a *SNR* of 5 dB as representative of real data. However, it is necessary to compare also results with different *SNRs*, assuming worst (0 dB) and best (10 dB) case scenarios.

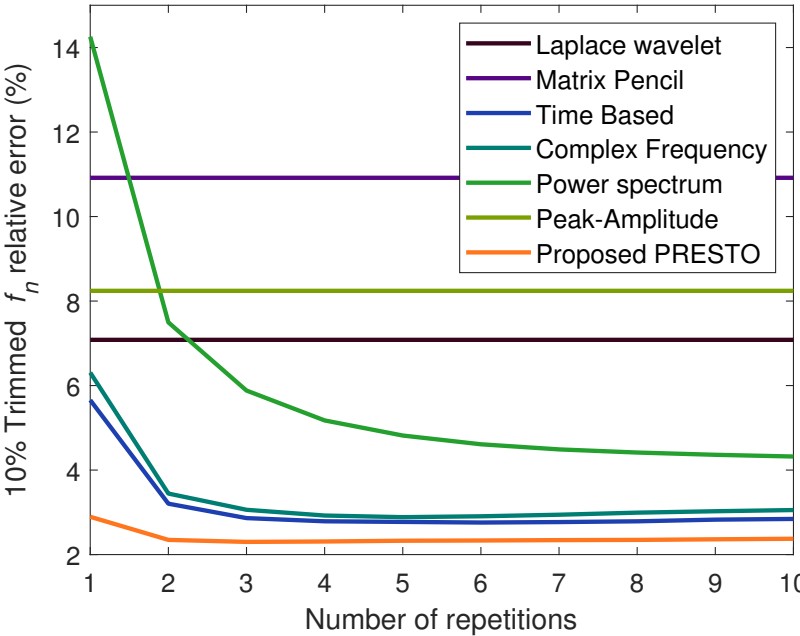

**Figure 11.** Average relative error (%) in the estimation of the natural frequency ($f_n = \omega_n/(2\pi)$) of two simulated underdamped modes excluding the worst 10% cases of the database. SNR 5 dB.

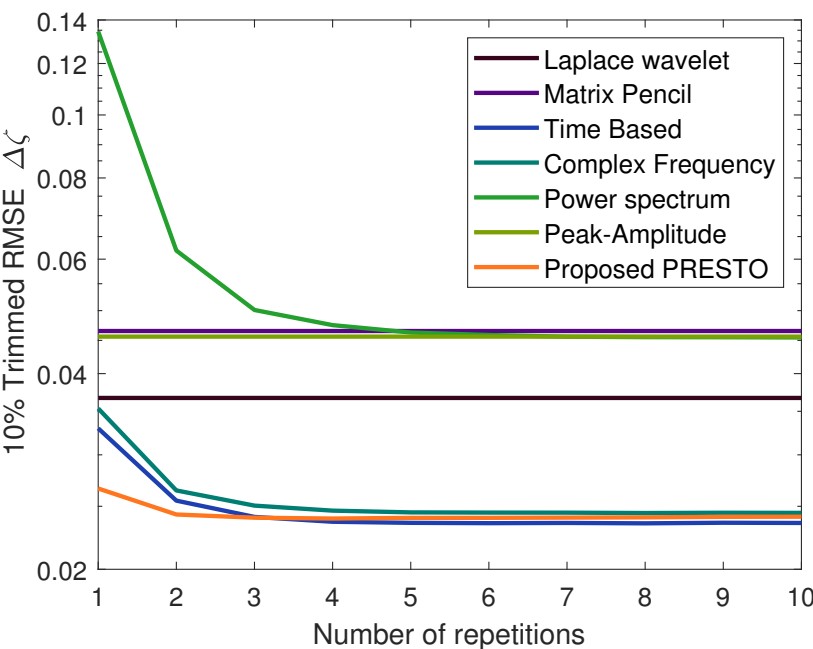

**Figure 12.** Root mean square error (RMSE) in the estimation of the damping factor ($\zeta$) of two simulated underdamped modes excluding the worst 10% of cases of the database. SNR 5 dB. Logarithmic scale.

**Table 1.** Main results of the simulated experiments comparing the average natural frequency error and damping factor RMSE, considering different SNRs.

|  | Time Based Estim. | Complex Freq. Estim. | Power Spec. Estim. | Matrix Pencil Estim. | Peak Amplitude Estim. | Laplace Wavelet Estim. | PRESTO Mixed Estim. |
|---|---|---|---|---|---|---|---|
| Num. of reps. | 6 | 6 | 10 | 1 | 1 | 1 | 2 |
| Time/est. | 4.1 s | 6.2 s | 25.6 s | 0.01 s | $5 \cdot 10^{-5}$ s | 0.06 s | 0.06 s |
| Avg. Err. $f_n$ SNR = 10 dB | 4.08% | 4.21% | 4.10% | 14.69% | 10.12% | 8.75% | 2.84% |
| Avg. Err. $f_n$ SNR = 5 dB | 6.41% | 6.78% | 6.16% | 14.69% | 10.17% | 8.78% | 4.37% |
| Avg. Err. $f_n$ SNR = 0 dB | 10.43% | 11.10% | 9.27% | 14.69% | 10.76% | 8.82% | 6.82% |
| RMSE $\zeta$ SNR = 10 dB | 0.029 | 0.030 | 0.042 | 0.095 | 0.049 | 0.042 | 0.026 |
| RMSE $\zeta$ SNR = 5 dB | 0.042 | 0.044 | 0.057 | 0.098 | 0.051 | 0.042 | 0.038 |
| RMSE $\zeta$ SNR = 0 dB | 0.061 | 0.067 | 0.078 | 0.095 | 0.061 | 0.043 | 0.055 |

For the results presented in Table 1, the performance analysis considered only 1 signal and the best number of repetitions of the optimization process for each technique, that is, 6 repetitions for the time based estimation and the complex frequency based estimation, 10 for the power spectrum based estimation, 2 for the PRESTO estimation and only 1 for Matrix pencil, Laplace Wavelet and Peak-Amplitude, since the paradigm is different for those last three techniques. All those started with initial random conditions for all the parameters but for the PRESTO estimation. In the case of the proposed PRESTO optimization, first 1 run of the Peak-Amplitude estimation was performed and then 2 more runs of the time based estimation. The computational cost was measured in processing time using a single 2.5 GHz CPU with 8 cores, and it measured the time of all the stages in the optimization process. In the case of the PRESTO solution we include both the time required to perform the Peak-Amplitude estimation and the time based estimation.

Regarding the errors in the estimation of the natural frequency considering different values for *SNR*, in all cases the PRESTO technique improved in approximately 1.5–2% the best estimation achieved by the next best technique. Considering an average case of 5 Hz

for $f_n$ and an average $SNR = 5$ dB, these results imply an average difference of less than 0.1 Hz between the real and the estimated frequencies.

When discussing the RMSE relative to damping factor, the results for PRESTO estimation are better in almost all cases. The only situation where the results were better by another technique was with Laplace Wavelet and $SNR = 0$. In this case, the Laplace Wavelet estimation threw an average error of 0.043 vs. 0.055, which is coherent with the statement that the technique is insensitive to noise.

In general, the proposed PRESTO estimation shows a very good response in terms of time and, more specifically, in terms of repeatability and accuracy. The technique is sensitive to noise and is only beaten compared to the Laplace Wavelet in terms of damping factor and under a noisy scenario. Notice that the Laplace Wavelet technique was already successfully employed on real F-18 analyses. For this reason, the Laplace Wavelet technique and the PRESTO estimation will be compared in processing real Flight Flutter data.

*6.2. Multimodal Estimation of Real In-Flight Responses*

6.2.1. Description of the Experiment Conditions

The Spanish Air Force Logistic Center of Armament and Experimentation (CLAEX as the Spanish acronym) has kindly provided the authors with several sets of sampled data from Flutter Flight Tests. The data was acquired on Spanish Air Force F-18A/B, flown from the Torrejon Air Force Base and controlled from the CLAEX Control Room. The aircraft configuration during the tests was cruise flight conditions, flaps and landing gear retracted and different stores hanging from inboard and outboard, wingtip stations loaded and centerline fuel tank. Stations 4 and 6 were empty.

The setup of the Flight Tests includes the use of an instrumented aircraft with extensometers to gather vibrational data and a Radio Frequency (RF) link with a ground station to direct the test. The excitations are typically made through the Flight Excitation Control Unit of the aircraft (FECU), which injects the predetermined excitation signals into the flaperons. The motion of the flaperons is then recorded through accelerometers in the trailing edge of the flaperons.

During the test, the test director is in continuous communication with the pilot from the control room and analyzes qualitatively the data in real time to avoid going beyond a catastrophic point. In case the test director determines that the signals analyzed imply a dangerous condition for the safety of the pilot, the Test Director commands to stop the test and the aircraft must return to known safe flight conditions, usually in the center of the envelope.

The data gathered during a Flutter Test typically includes Sine Dwell excitations and Frequency Sweep excitations. All those excitations are regular data extraction techniques for the F-18 legacy aircraft series, as described in [1,34]. However, the data points selected in the present paper are limited to Sine Dwell excitations.

The expectations of the test are to generate a flight envelope for a given configuration of the aircraft and subconfigurations, where only the most critical subconfigurations will be tested and the rest are limited by the most restrictive conditions of the former. In particular, the Flutter Flight Tests aim to define the combination of maximum Mach number at different altitudes available for the aircraft in safe flight conditions. EASA CS-25 and MIL-A-8870 [5,6] specify a minimum value for reduced structural damping factor of 0.03 (equivalent viscous damping factor of 0.015) as the limiting flutter point to stop the test and reach the edge of the envelope. Values of damping factor below 0.03 are considered to imply flutter conditions, and therefore those regions must be avoided and documented.

The signals intended to be analyzed are the response of the aircraft after an excitation at a given frequency, represented in Figure 4 as the relax phase of the extensometers, with no input from the flaperons (no excitation). As we can see, Equation (1) describes the natural vibration of an element of a lifting surface of an aircraft in generalized coordinates, in a frequency region where a single mode can be isolated. It can be interpreted, for example, as the angle registered by an inclinometer on the leading edge of the wing. If for

any reason the wing absorbs energy at frequency $f_n$, it will return that energy oscillating as stated in Equation (1) when the excitation stops.

Please note here that, even though one could assume that the phase of the mode might be neglected by properly aligning the time window with the signal, this will not be the case when multiple modes are present in the signal. When two different modes contribute to the signal they can compensate each other, so that even though the signal might start in a zero value, both modes can present non zero phase. This assumption can be explained during the acquisition process. When the excitation stops (see Section 2), the FECU unit stops the flaperons at $0°$, which in the case of a single mode contributing and relatively far from the resonance frequency (most of the cases analyzed), often coincides with the measuring extensometers in a relaxed state (offsets of $0°$ or $180°$), hence returning a zero (or close to zero) value for the starting point when the data are acquired, and indirectly setting the contributing mode phase angle to a value close to $0°$ or $180°$. In the case of $180°$, the amplitude constant fixes the issue by returning a negative constant. In the case of multiple modes, however, the contribution of both modes implies that none of the modes need to be zero if the contribution of both modes add up to zero or close enough to zero. In fact, when two modes contribute the oscillation is seldom aligned in phase with the excitation.

Data was acquired from Sine Dwell excitations at 85 Hz, with a trimmed maximum duration of 5 s each set. Shorter sets were zero-padded in the final samples to process samples of the same length. Take into consideration that not all the data was usable, since many sets included turbulence, unstable maneuvers, etc. In addition, the data provided did not include the actual values for frequency, damping factor or phase, so it was necessary to estimate the values by comparing the solution with the real data, without any previous estimation on the true data. On top of that, one important point to remark is the Sine Dwell excitations were run in batches of 8 different programs, meaning that (as a simplified example) if the expected natural frequency lies at say 4.5 Hz, the excitation programs started at 4.2 Hz, 4.3 Hz and so on until 4.9 Hz (eight programs in total). This fact is crucial for the upcoming analysis since all the signals, 640 in total, were processed independently of their closeness to the actual natural mode, and therefore assuming a perfect noiseless scenario, where all the data were accurately gathered and each single signal had good data, only 1/8 to 3/8 of the signals shall include any kind of useful data, given that those are the closest to the natural frequency of interest. In our case this means that the best expectation from the analysis would be to get 80–240 good data samples assuming that all the data gathered are good and usable, which is an assumption that is seldom correct.

As a summary, the ultimate goal of this problem will be to fit real inflight data with the two main parameters of this model, $f_n$ and $\zeta$, for each critical natural mode and at every dynamic pressure.

### 6.2.2. Analysis of Results

Figure 13 depicts the plot of original data amplitudes vs. synthetic data amplitudes considering the proposed PRESTO technique. The optimization assessment was made representing all the individual data samples amplitudes vs. the signals reconstruction. Considering this approach, a perfect fit would return a perfect straight line at $45°$ ($slope = 1$) with a $y - intercept$ of $y = 0$. Out of 640 total datasets, the best 180 were represented.

The numerical results of the optimization considering the phase angle can be seen in Table 2. Calculated over the 180 datasets, show an R-Square coefficient of 0.72, which implies a small amount of dispersion of the data (an ideal R-square value of 1 would imply a perfect fit). On top of that the slope of the regression is 0.83, close to the ideal slope of 1, and a $y$-intercept of $-0.001$.

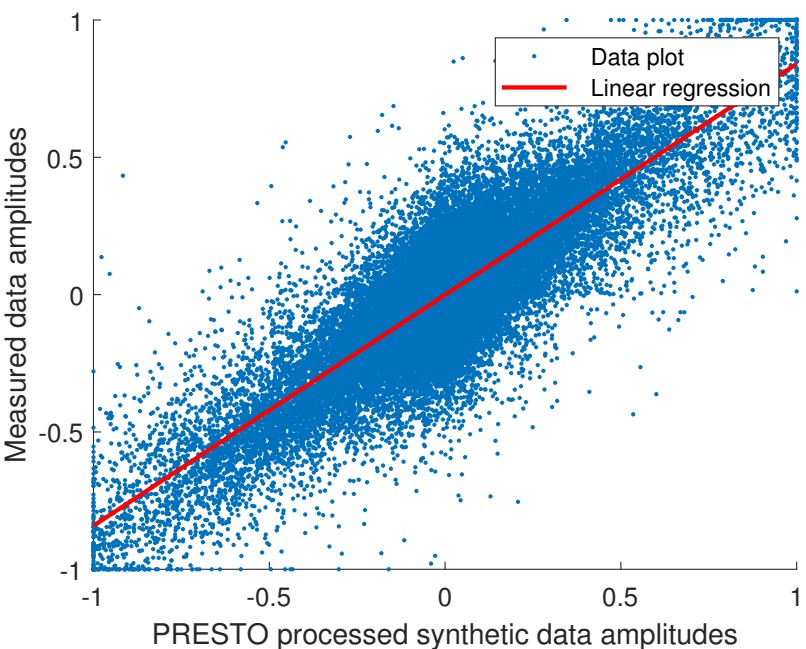

**Figure 13.** Data fit of real vs. synthetic data processed with the PRESTO technique calculating both modes simultaneously.

Comparing the PRESTO with the Laplace Wavelet estimation and regressions, Figure 14 and Table 3 represent the same results but without considering the phase of the spectrum in the fitting calculations. In this case, the plot shows also a clear correlation in the data as in the case of the PRESTO estimation. However, the R-Squared coefficient is slightly worse than in the case of the PRESTO estimation (0.57) and a very similar slope (0.79), which implies that the fitting results are mostly equally coherent in both cases, although the Laplace Wavelet estimation shows slightly more dispersion in the data than the PRESTO technique.

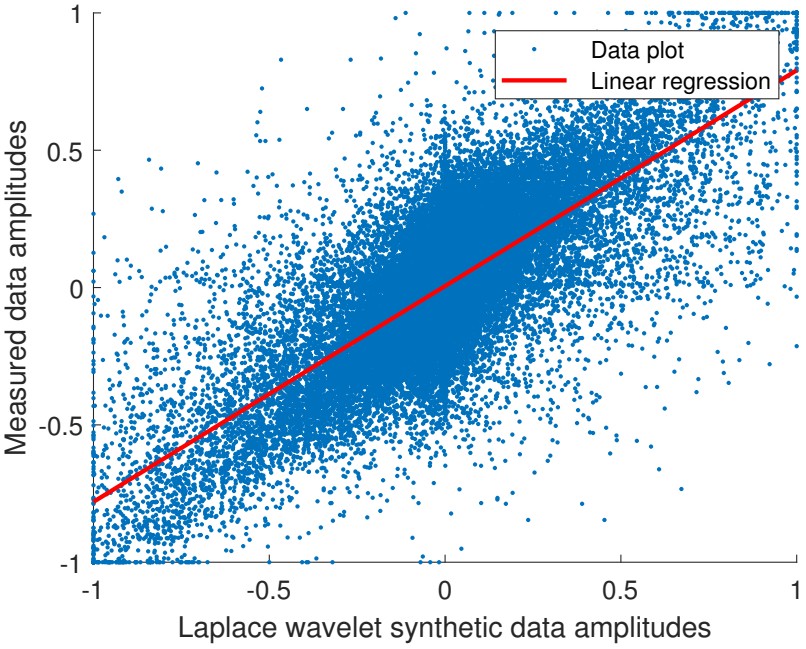

**Figure 14.** Data fit of real vs. synthetic data processed with the Laplace Wavelet technique calculating one mode at a time.

**Table 2.** Linear regression over sampled data vs. reconstructed data considering the proposed PRESTO estimation.

| Linear Regression Equation | | $y = Mx + N$ | |
| --- | --- | --- | --- |
| | Nominal value | Lower bound | Upper bound |
| $M$ | 0.8302 | 0.8264 | 0.8339 |
| $N$ | −0.0007 | −0.0015 | 0.0001 |
| R-Squared coefficient | | 0.716 | |

**Table 3.** Linear regression over sampled data vs. reconstructed data considering the Laplace Wavelet estimation.

| Linear Regression Equation | | $y = Mx + N$ | |
| --- | --- | --- | --- |
| | Nominal value | Lower bound | Upper bound |
| $M$ | 0.7856 | 0.7809 | 0.7903 |
| $N$ | 0.0055 | 0.0045 | 0.0064 |
| R-Squared coefficient | | 0.572 | |

Figures 15–18 show the results of the match between original and synthetic reconstructions of selected signals, PRESTO processing vs. Laplace Wavelet processing. The selection has been made under the following rules:

- Coupled plots (horizontally) represent the same original signal processed by the PRESTO technique (left) and Laplace Wavelet (right).
- The signals have been sorted by the MSE between the original and reconstructed signals, and obviously the MSE order is different for Laplace and PRESTO processing. Order 1 means the lowest MSE obtained during the reconstruction.

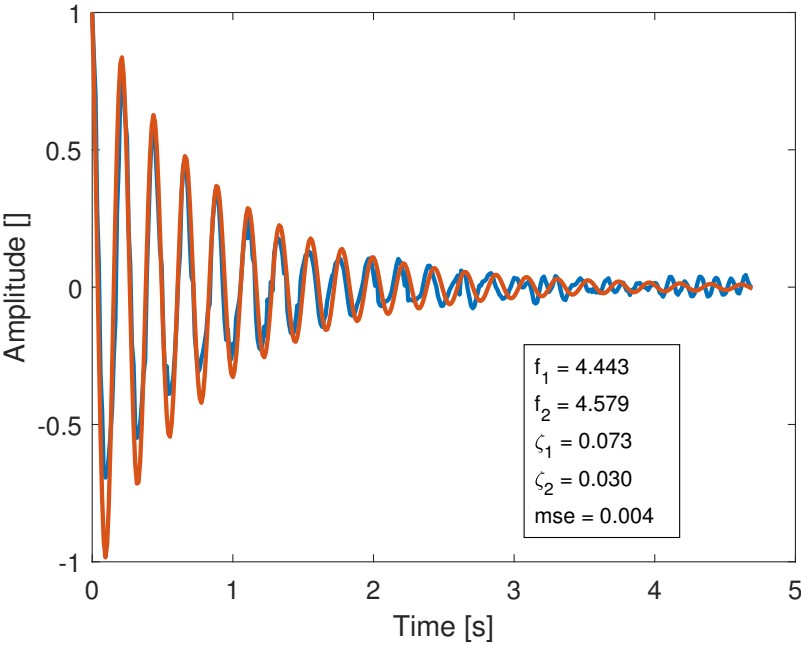

**Figure 15.** Original signal (blue) and signal reconstruction processed by the PRESTO technique (red). MSE order 7 (lower order means better match).

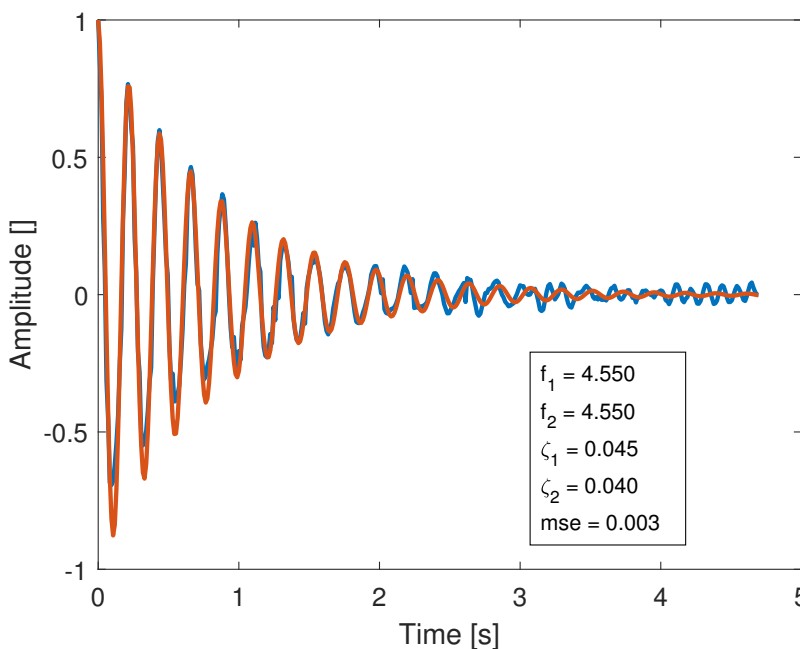

**Figure 16.** Original signal (blue) and signal reconstruction processed by the Laplace Wavelet technique (red). MSE order 1 (lower order means better match).

Figures 15 and 16 represent the matching for one of the signals where only 1 mode is involved, while Table 4 shows the estimated parameters after forcing the algorithm to extract two different modes. In this case the Laplace processing returns a very good match. In fact it is the best of the whole dataset processed by that technique, while the PRESTO processing returns a slightly worse error (the 7th from the whole dataset). In this case, the Laplace Wavelet processing returned two equivalent modes ($f$ = 4.5 Hz, $\zeta$ = 0.04), while the PRESTO processing returned one first extra mode disturbing the signal and clearly spurious, while the second mode of $f$ = 4.6 Hz, $\zeta$ = 0.03 is clearly dominant (but disturbed by the former). After processing the signal through logarithmic decrements, the results show a damping factor of $\zeta_{declog}$ = 0.039, which is much closer to the Laplace Wavelet processing. The explanation to the better estimation in damping for the Laplace wavelet is described marginally in Section 6.2.3. The PRESTO mechanism forces the existence of two different modes regardless of the actual number of modes involved, while the Laplace Wavelet will try to match one single mode with the maximum energy and then start searching for secondary modes. This means that if only one mode is involved, the PRESTO algorithm will initially try to share the energy between two "ghost" modes, neither of them specially accurate but close to the actual value, while the Laplace Wavelet will allocate the maximum amount of energy into one single mode (initially the actual mode) and if forced to search more modes those will be spurious modes. According to the authors experience analyzing real data, the most sensitive factor is damping. While the frequency is often estimated within an acceptable range of values, the damping factor is more sensitive to local minima or oscillations in the estimations. That being considered, in Figures 17 and 18 the effect is the opposite. The PRESTO algorithm replicates very accurately the features of the original signal including artifacts (Figure 17), which gives a slight advantage on the confidence on the results compared to the Laplace Wavelet (Figure 18), always noting that the results of the Laplace Wavelet replicate with high accuracy the original signal. In this case, however, it is not possible to perform an even-breaker test. The Logarithmic decrements technique is not applicable for this signal, and therefore the only metric capable of giving confidence is the one exposed above.

**Table 4.** Figures 15 and 16 estimated parameters.

| | Figure 15 (PRESTO) | | Figure 16 (Laplace) | |
|---|---|---|---|---|
| | Frequency [Hz] | Damping factor [] | Frequency [Hz] | Damping factor [] |
| *Mode* 1 | 4.443 | 0.073 | 4.550 | 0.045 |
| *Mode* 2 | 4.579 | 0.030 | 4.550 | 0.040 |
| MSE | 0.004 | | 0.003 | |

Figures 17 and 18 show results where clearly two modes are involved, and as in the previous case, Table 5 shows the estimated parameters for those two modes. In this case the PRESTO processing order is the 10th from the whole dataset, while the Laplace Wavelet processing returned the 22nd in order. Where two modes are involved it is not possible to apply logarithmic decrements to the data, and unfortunately the authors are unable to compare both results to a tie breaker. Nevertheless, notice that the matching of the PRESTO processing mimics almost perfectly the artifacts in the signal, while the Laplace Wavelet processing fails to match slightly in amplitude and to reproduce the artifacts, which is coherent with the analysis on the techniques performed in Section 6.2.3.

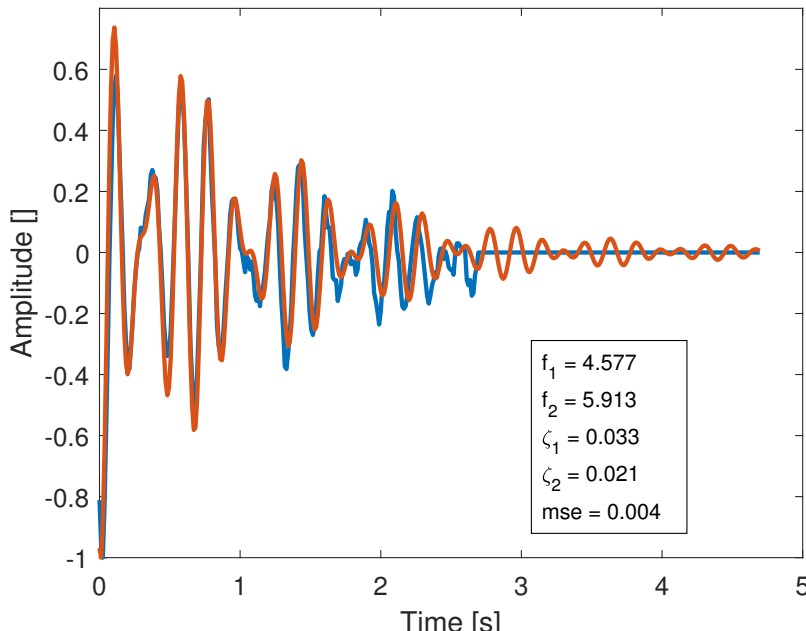

$$f_1 = 4.577$$
$$f_2 = 5.913$$
$$\zeta_1 = 0.033$$
$$\zeta_2 = 0.021$$
$$mse = 0.004$$

**Figure 17.** Original signal (blue) and signal reconstruction processed by the PRESTO technique (red). MSE order 10 (lower order means better match).

**Table 5.** Figures 17 and 18 estimated parameters.

| | Figure 17 (PRESTO) | | Figure 18 (Laplace) | |
|---|---|---|---|---|
| | Frequency [Hz] | Damping factor [] | Frequency [Hz] | Damping factor [] |
| *Mode* 1 | 4.577 | 0.033 | 4.600 | 0.040 |
| *Mode* 2 | 5.913 | 0.021 | 6.000 | 0.030 |
| MSE | 0.004 | | 0.006 | |

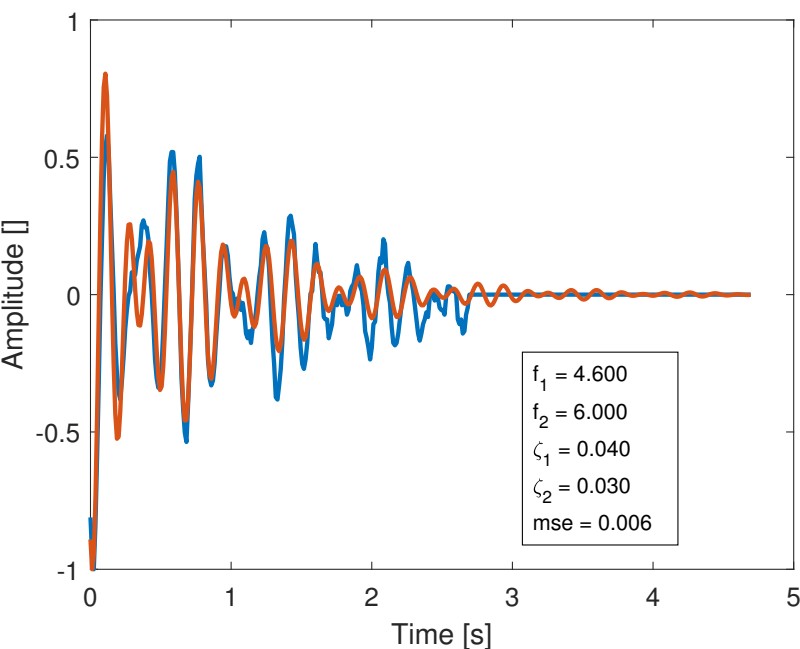

**Figure 18.** Original signal (blue) and signal reconstruction processed by the Laplace Wavelet technique (red). MSE order 22 (lower order means better match).

### 6.2.3. Comparison between PRESTO and Laplace Wavelet Techniques

Laplace Wavelet and PRESTO techniques are quite similar. Considering one single degree of freedom for simplicity, both equations are equivalent as shown in Equation (29)

$$Ae^{\omega_n 2\pi(t-\tau)}sin(\omega_d 2\pi(t-\tau)) = Ae^{\omega_n 2\pi t - \omega_n 2\pi\tau}sin(\omega_d 2\pi t - \omega_d 2\pi\tau) = Be^{\omega_n 2\pi t}sin(\omega_d 2\pi t - \phi) \tag{29}$$

This result can be extrapolated for 2 degrees of freedom if instead of 1 single time delay 2 different time delays are considered, so all in all both equations are exactly the same in terms of unknown parameters. However, it is important to notice that both techniques differ in the method of process data. Laplace Wavelet technique estimates one mode at a time, removes its contribution to the original signal and afterwards estimates the second mode. Errors in the estimation of the first mode will be carried and amplified for the estimation of the second mode, mainly driven by errors in the phase angle as demonstrated in [44] for Matching Pursuit estimations, which is the search procedure followed by the Laplace Wavelet approach. This problem is partially mitigated by the PRESTO estimation. The estimation of both modes at the same time implies that the error is not carried to the second mode, since the optimization intends to minimize the error contribution of both modes globally. This fact might explain why the Laplace Wavelet fits considerably better signals where the contribution of the second mode is negligible. Figures 16 and 18 show this behavior, where the estimated signal is affected by an offset. Take into account that the addition of a second mode sensibly not aligned with the corresponding original signal will induce a slight offset in the sum signal.

It may be possible to modify the technique to calculate both modes at the same time through the Laplace Wavelet but take into account that Matching Pursuit is a greedy estimator [39,40], and the memory required to construct the dictionary is comprised by the product of the number of elements of each parameter. Consider for example a typical range in frequency between 2 Hz and 10 Hz and a low resolution of 0.1 Hz. There are 81 elements in this range. Assuming all the parameters around this number of elements, with only 1 mode at a time (4 parameters), the dictionary size is $81^4 \approx 4 \cdot 10^7$ elements, with their respective calculations associated. In the case of 2 modes (8 parameters) the dictionary size would be $81^8 \approx 2 \cdot 10^{15}$ elements. Even assuming the test facility has a mid size cluster

available to calculate low accuracy results, if two modes are to be processed simultaneously with the Laplace wavelet approach, it is not easy to get a real time estimation during a Flutter Flight Test as it may be possible using the PRESTO estimation, even using a low end home use computer and reaching more accurate results.

## 7. Conclusions and Discussion

The present paper has developed the equations of motion of a lifting surface subjected to aerodynamic, inertial and elastic forces, with damping factor defined by the viscous damping model. It has shown the importance of the phase angle in relationship with the bandwidth and the resonance frequency and presented a new approach to process data, the PRESTO technique, comparing it to a state-of-the-art technique, the Laplace Wavelet processing.

The main conclusion to be drawn from this paper is that the phase angle contribution is paramount for the identification of natural modes in flutter data processing (or in general in aeroelasticity), either as a phase angle like in the PRESTO technique or as a time offset as in the Laplace Wavelet approach as demonstrated in Section 6. On top of that, a new data processing technique has been presented, the PRESTO technique. The validation of such processing approach has been demonstrated by the numerical results with two different sets of data, simulated data (generated with known parameters) and real data (without any previous knowledge of the actual values), and it has demonstrated to be a robust and accurate technique to identify natural modes of Flutter Flight Tests, especially comparing it to the already in use technique, Laplace Wavelet. It is worth noticing that the results between both techniques, Laplace Wavelet and PRESTO, are quite similar. In addition, it must be taken into account that the Laplace Wavelet technique relies upon Matching Pursuit processing. Given the greedy nature of Matching Pursuit, the best combination of parameters will be included as a solution of the Laplace Wavelet only if the model matches reality as accurately as possible (it was already demonstrated that the Laplace Wavelet fitting function does) and the dictionary includes the right combination of parameters. However, increasing the number of parameters (or the size of the dictionary) increases exponentially the number of cases to calculate and consequently the time required, which in some cases might lead to extremely long calculation times if the seeds and ranges are not estimated carefully.

The PRESTO and Laplace Wavelet processing show the same level of correlation between the reconstructed and original signals, although the Laplace Wavelet shows slightly more dispersion and mimics artifacts worse. The dictionary conditions enormously the performance of the technique, and therefore the Laplace Wavelet performance will be sensibly worse considering a non-taylored dictionary. Regarding the robustness, the Laplace Wavelet shows an extreme insensibility to noise and overfitting when less modes than expected are considered compared to the PRESTO technique. However, the Laplace Wavelet technique has difficulties processing signals consisting of more than two modes, while the PRESTO technique has a tendency to overfitting signals consisting of one single mode.

In general, PRESTO and Laplace Wavelet approaches show the same level of correlation between the original and reconstructed signals. While the PRESTO technique showed superior characteristics than the reference technique analyzed, Laplace Wavelet, in terms of performance, dispersion and several modes processing, the Laplace Wavelet shows stronger robustness in terms of overfitting of signals consisting of one single mode and noise.

As for the recommendations and future lines of work, the PRESTO technique shows great potential as a stand-alone technique for real time applications, since the model accurately matches reality and the time required is short enough as to allow for getting results before proceeding with the next test point. However, there is some room for improvement. The technique is prone to overfitting, and simple unimodal signals will return bigger errors than the Laplace Wavelet or even logarithmic decrements. A possible line of work may be to develop a preprocessing algorithm to estimate the most probable

number of modes, which will significantly reduce (or even eliminate) the propensity to overfitting. In addition, it is necessary to improve its robustness to noise without getting stuck in local minima. For that reason it is necessary to improve the optimization algorithm. At last, the datasets employed came from F-18 real flights. The technique may be most likely extended to different platforms, but it needs to be validated with real data from other aircraft.

**Author Contributions:** Conceptualization: S.A.-K., R.G.-P. and M.R.-Z.; Formal analysis: R.G.-P. and S.A.-K.; Funding acquisition: M.R.-Z. and R.G.-P.; Investigation: S.A.-K. and R.G.-P.; Methodology: S.A.-K. and R.G.-P.; Project administration: M.R.-Z. and R.G.-P.; Software: R.G.-P. and S.A.-K.; Supervision: M.R.-Z. and R.G.-P.; Validation: S.A.-K. and R.G.-P.; Visualization: S.A.-K. and R.G.-P.; Writing; Original draft: S.A.-K.; Writing; Review and editing: M.R.-Z. and R.G.-P. All authors have read and agreed to the published version of the manuscript.

**Funding:** This work has been partially funded by the Spanish Ministry of Economy, Industry and Competitiveness, with project RTC-2016-4687-7, the Spanish Ministry of Science, Innovation and Universities, with project RTI2018-098085-B-C42 (MSIU/FEDER) and by the Community of Madrid and University of Alcala under project EPU-INV/2020/003.

**Data Availability Statement:** Restrictions apply to the availability of these data. Data were obtained from the Spanish Air Force CLAEX, with permission to use and report in the present paper (under restrictions) from the Communications Office of the Spanish Air Force. The data are not publicly available for confidentiality reasons.

**Acknowledgments:** The data from real Flight Flutter Tests were kindly provided by the Spanish Air Force, the institution to which the authors show their appreciation. The plots in Appendix A were extracted from two different software packages developed by Agustin Vaquero for the Spanish Air Force CLAEX.

**Conflicts of Interest:** The authors declare no conflict of interest. The funders had no role in the design of the study; in the collection, analyses, or interpretation of data; in the writing of the manuscript, or in the decision to publish the results.

## Appendix A. Support Figures

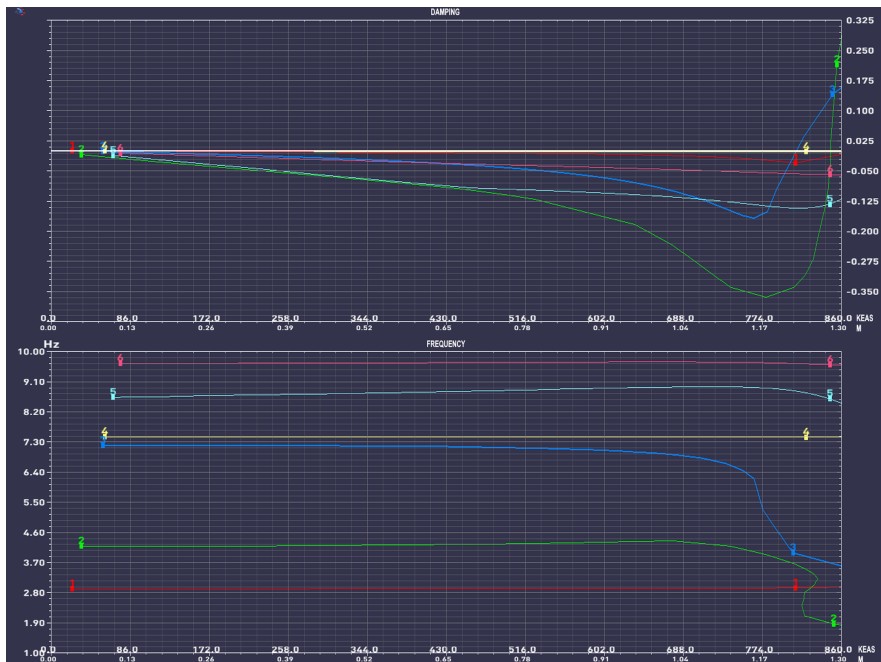

**Figure A1.** Typical V-G sample plot from VGPlotter. Notice the frequency variation in Modes 2 and 3 and the change from negative to positive damping factor (flutter point) in the vicinity of the airspeeds where those two frequencies get close together.

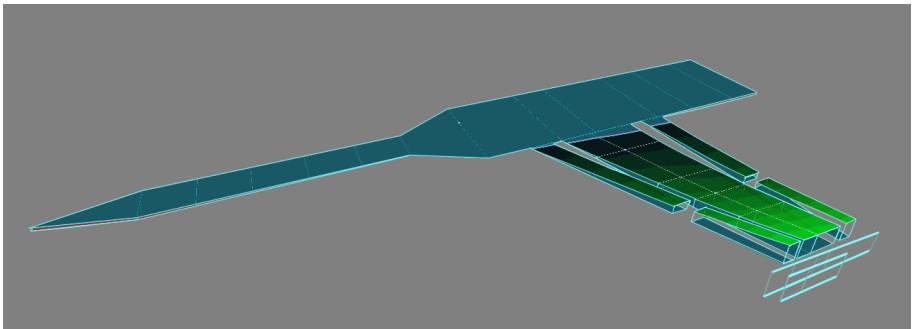

**Figure A2.** Structural deformation sample depiction from DePlotter. The figure shows a typical pure wing plunge mode, with a subtle pitch deformation in the fuselage.

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
