# Peer review of "Multimodal Estimation of Sine Dwell Vibrational Responses from Aeroelastic Flutter Flight Tests"

_aerospace, doi:10.3390/aerospace8110325_

Round 1
Reviewer 1 Report
The paper has been very interesting for me. Excellent job.

Author Response
Dear reviewer.
Thank you kindly for your touching and encouraging words of support, along of course with the endorsement of our paper. We deeply appreciate such warming review.
In case you can´t access the Editor´s comments we provide you with a copy (attached), stressing once again our appreciation for your effort and kind words of support and reflected also in the letter to the Editor.
Thank you dearly.
Sami.

Reviewer 2 Report
General Comment
The authors developed and compare a method for estimating modal parameters in flutter flight tests, the PRESTO. I found the paper interesting and covering an active research topic. However, from my point of view there are some issues that should be addressed before the paper is suitable for publication.
Comments and Questions:
On page 1, where it reads “short time data imply”, it should read “short time data implies”.
There are no references in the paper, only their numbering in the text.
From my point of view, Figures 16, 17 and 18 should be in the main text.
On page 1, where it reads “extensive previous analysis must be performed”, it should read “extensive previous analyses must be performed”.
Also on page 1, where it reads “An F-18”, it should read “A F-18”.
On page 2, where it reads “those analysis”, it should read “that analysis”.
On page 3, where you describe how the paper is organized, it misses mentioning section 5.
Throughout the text, where you use the term “damping” to quantify damping, from my point of view, you should use “damping factor” instead.
In Figure 2, in the “Structural Model” box, you should replace “Rigidity Matrix” for “Stiffness Matrix”.
What happens in the case the aerodynamics are not accurately estimated and it has resulted in flutter sooner than expected? From Figure 2 you would choose path 11 because it has led to flutter, but the aerodynamic model is not accurate and consequently neither the aeroelastic one.
At the end of path 4 you mention that the natural frequencies, mode shapes and damping ratios of the structure are obtained without the aerodynamic forces. Wouldn’t it be more accurate if you account for these aerodynamic forces (i.e. accounting for flexibility effects)?
On page 6, where it reads “omega” you should use its Greek symbol used in the equations instead.
On page 8, where it reads “to analyze to assess”, it should read “to assess”.
Some of the acronyms haven’t been introduced, e.g. SVD.
Which are the stopping criteria defined for the optimization reported on page 11?
Why have you chosen the trust region algorithm? Does it perform better for this kind of optimization problems?
On page 14, where it reads “Timely-wise”, it should “Timewise”.
The differences between methods for frequencies are small, although the same cannot be said for the damping ratios. Can you elaborate further on this outcome?
Furthermore, a table summarizing the data from Figures 12 to 15 with the relative differences between frequencies and damping ratios from different methods would improve the paper.
On page 19, where it reads “in general aeroelastic”, it should read “in general aeroelasticity”.
Author Response
Dear reviewer.
Thank you kindly for your review. We acknowledge your comments are important notes that, in several cases should have been captured by us (we apologize and show our appreciation on your timely effort), and in some others will significantly improve the quality of the paper (for those of course we also appreciate your time and effort).
All your comments have been incorporated to the current version of the paper. As a general note, you will see the most substantial comments in red in the text, so that you can pinpoint them easily (typographical errors were simply fixed without changing color). If you may kindly provide some more of your time, you will find a brief explanation on the rationale behind the changes in the Editor's letter (attached), since we believe the format will be easier to read.
At last, than you dearly again for your time and effort. The comments are not trivial. Show a deep concern about the quality of our paper, many hours of hard work to provide us such detailed and accurate comments and demonstrate many years of expertise analyzing and studying flutter.
Sami.

Round 2
Reviewer 2 Report
The authors have addressed my concerns.